# Search Self-play: Pushing the Frontier of Agent Capability without Supervision

Hongliang Lu[1,2]*, Yuhang Wen[1,3]*, Pengyu Cheng[1]†, Ruijin Ding[1], Jiaqi Guo[1], Haotian Xu[1], Chutian Wang[1], Haonan Chen[1], Xiaoxi Jiang[1], Guanjun Jiang[1]

[1] Qwen Large Model Application Team, Alibaba
[2] Peking University
[3] Sun Yat-sen University

## Abstract

Reinforcement learning with verifiable rewards (RLVR) has become the main-stream technique for training LLM agents. However, RLVR highly depends on well-crafted task queries and corresponding ground-truth answers to provide accurate rewards, which requires significant human effort and hinders the scaling of RL processes, especially in agentic scenarios. Although a few recent works explore task synthesis methods, the difficulty of generated agentic tasks can hardly be controlled to provide effective RL training advantages. To achieve agentic RLVR with higher scalability, we explore self-play training for deep search agents, in which the learning LLM utilizes multi-turn search engine calling and acts simultaneously as both a task proposer and a problem solver. The task proposer aims to generate deep search queries with well-defined ground-truth answers and increasing task difficulty. The problem solver tries to handle the generated search queries and output the correct answer predictions. To ensure that each generated search query has accurate ground truth, we collect all the searching results from the proposer's trajectory as external knowledge, then conduct retrieval-augmentation generation (RAG) to test whether the proposed query can be correctly answered with all necessary search documents provided. In this search self-play (SSP) game, the proposer and the solver co-evolve their agent capabilities through both competition and cooperation. With substantial experimental results, we find that SSP can significantly improve search agents' performance uniformly on various benchmarks without any supervision under both from-scratch and continuous RL training setups. The code is at `https://github.com/Qwen-Applications/SSP`.

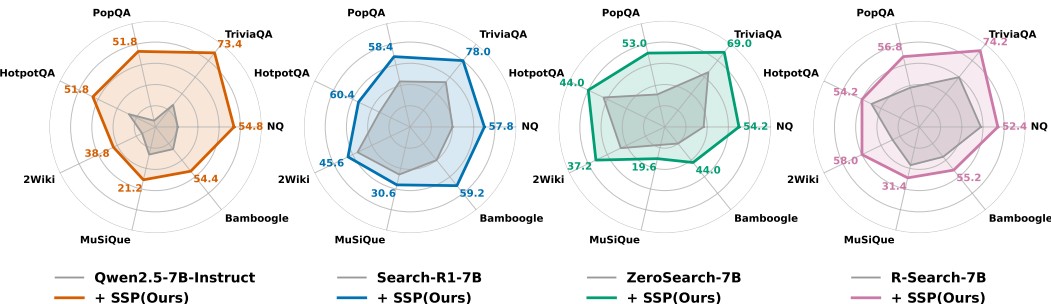

Figure 1: Performance gains of deep search agents trained via Search Self-play (SSP) across various agentic benchmarks. Our SSP method uniformly surpasses multiple strong open-source baselines without any agentic data annotation and additional supervision.

---

*Equal contribution, work done during an internship at Alibaba.
†Corresponding author.

## 1 INTRODUCTION

The rapid development of large language models (LLMs) has enabled Artificial Intelligence (AI) with astonishing natural language capabilities and systemically reshaped various application scenarios, including machine translation (Brown et al., 2020; Zhang et al., 2023), dialog systems (Ouyang et al., 2022; Achiam et al., 2023; DeepSeek-AI et al., 2025), document retrieval (Zhu et al., 2023), and AI search (OpenAI, 2025; Li et al., 2025c). In this revolution, AI agents, which utilize LLMs' power to interact with complex functional tools and solve multi-step decision-making processes, have attracted wide attention for their unprecedented application potential and commercial value (Li et al., 2024b; Xi et al., 2025b). According to different available tool sets, LLM Agents can be further categorized. For example, deep search agents primarily use search engines (Jin et al., 2025b; Team, 2025), GUI agents are multi-modal based on screenshots (Wang et al., 2024; Xie et al., 2024; Wang et al., 2025), and coding agents utilize code interpreters (Huang et al., 2023; Yang et al., 2024b).

Although with great practical potential, training LLM Agents has been widely acknowledged as a challenging task due to the scarcity of supervised training data (Qi et al., 2024; Team et al., 2025). Besides, different types of LLM agents utilize diversified tool sets, so the strategies of agents with different tool sets can be entirely dissimilar. Even for the same task query, a human-annotated agentic trajectory can be inapplicable for another agent tool set, which further exacerbates the data shortage. Thanks to the breakthroughs of reinforcement learning with verifiable rewards (RLVR) (Guo et al., 2025a), many recent works start to train LLM agents within the RL paradigm instead of being stubborn in the supervised data collection (Shang et al., 2025; Jin et al., 2025b). In these agentic RL methods, a ground-truth answer is well-crafted for each given task query, and the reward outcome is simply to check whether the agent's predicted answer is equivalent with the ground-truth. Agentic RL methods only concern about whether the final prediction is correct, without imposing any restrictions on the intermediate multi-step agents' exploration, which significantly reduces the demands of manual annotation (Zhang et al., 2025). However, agentic RLVR still heavily depends on a large amount of well-crafted verified ground-truth for training scaling-up (Zhao et al., 2025a; Zhang et al., 2025), which means the data scarcity problem remains a bottleneck for effective agentic training.

To further mitigate the annotation scarcity of agentic RLVR, query-synthesis methods have been explored (Li et al., 2025b; Gao et al., 2025a). Provided with a ground-truth answer, synthetic methods first select a simple question or a related condition, then recursively replace some of the key information from the question or the condition with more complicated descriptions. With this multi-step inject-then-fuzz data pipeline, one can generate agent queries with different multi-hop conditions with controllable task difficulty (Li et al., 2025a). However, query-synthesis approaches still suffer from two critical limitations of training efficiency and effectiveness: First, the training scalability is inherently constrained, as each synthesized question-answer pair must be rigorously validated for answer correctness and logic consistency to compute accurate task outcomes (Villalobos et al., 2024). Second, the offline synthetic scheme lacks the adaptability to dynamically adjust question difficulty to provide effective advantages during the RL training (Guo et al., 2025b). Consequently, existing approaches remain unqualified to be scalable and self-sustaining to generate high-quality agentic question-answer pairs without human annotation.

On the other hand, self-play methods, pioneered by AlphaGo Zero (Silver et al., 2017), have shown their effectiveness to continuously improve the intelligence of agents by playing games against agents themselves (Schrittwieser et al., 2020; Zha et al., 2021). With a well-defined gaming outcome computation, self-play methods collect different trajectories from both winner and loser, then reinforce the policy models without any additional supervision. Recent studies have also verified the effectiveness of self-play training for LLMs, specifically in improving safety (Deng et al., 2025a; Liu et al., 2025b), alignment (Chen et al., 2024; Cheng et al., 2024b) and reasoning (Cheng et al., 2024a; Chen et al., 2025a). Although self-play is naturally a potential solution to address data scarcity, its application for agent training remains unexplored.

To address the annotation scarcity of agentic RL training, this paper targets exploring the self-play training to self-improve agents under the deep search scenarios. More specifically, we design a Search Self-play (SSP) game, in which the target LLM simultaneously plays two alternating roles: a *question proposer* and a *problem solver*. The proposer generates deep search queries with verifiable ground-truth with progressive difficulty, while the solver attempts to answer the generated questions via multi-turn reasoning and search calling. To validate the correctness of each generated query, we collect all the searching results from the proposer's trajectory as the external materials, then

conduct a retrieval augmentation generation (RAG) to check if the solver can successfully predict the answer with all necessary information provided. With the above design, the deep search agent can autonomously generate high-quality training tasks and then solve them by itself, removing the demands of human-annotated verification while maintaining reward accuracy. Besides, the difficulty of the training queries becomes adaptive by controlling the reinforcement level of the proposer based on its SSP win rates. Through competition and collaboration in SSP, the proposer and solver co-evolve, systematically improving the target LLM's capacities of searching, reasoning, and self-verification. In experiments, we show that SSP yields substantial and consistent improvements across various benchmarks under both from-scratch and continual learning setups, establishing a scalable pathway toward self-supervised agentic training.

## 2 RELATED WORK

### 2.1 DEEP SEARCH AGENTS

Deep search agents leverage the power of search engines and the reasoning capacities of LLMs to conduct multi-turn retrievals and analyses for seeking accurate answers of complex and challenging questions, which have gained increasing attention for their huge application potential to serve people as a novel information acquisition paradigm (Huang et al., 2025b; Xi et al., 2025a). In contrast to traditional Retrieval-Augmented Generation (RAG) methods (Lewis et al., 2020), deep search agents employ multi-hop reasoning, dynamic query reformulation, and self-guided exploration to emulate a human-like investigative process (Xi et al., 2025a), which is crucial for applications that demand high precision and traceability, *e.g.*, scientific literature review (OpenAI, 2025), legal analysis (Li et al., 2024a), and fact-checking (Wei et al., 2024). Proprietary agents such as DeepResearch (OpenAI, 2025), Grok-3 (x.ai, 2025), and Kimi-Researcher (Moonshot AI, 2025) have already demonstrated noticeable performance on complicated information-seeking tasks. However, their model designs and training details remain opaque. In contrast, open-source efforts such as Search-R1 (Jin et al., 2025b), R1-Searcher (Song et al., 2025), DeepResearcher (Zheng et al., 2025), and ZeroSearch (Sun et al., 2025a) leverage agentic reinforcement learning (RL) to enhance question-answering capabilities, yet are still constrained by a limited amount of training queries. To scale up agentic RL, recent works such as WebDancer (Wu et al., 2025), WebSailor (Li et al., 2025b), and ASearcher (Gao et al., 2025b) propose question-synthesis pipelines. However, their processes remain offline, which are incapable of adaptively controlling task difficulty for providing more effective RL advantages. We propose a self-play search agentic training scheme, in which the learning agent generates tasks and then solves them simultaneously. Through self-play training, the agent's task-proposing and problem-solving abilities co-evolve without supervision, which significantly reduces human annotation and extends the agentic training to broader scenarios.

### 2.2 SELF-PLAY IN LARGE LANGUAGE MODELS

Self-play methods let the target model play different roles in a multi-agent system, then update the policy with collected agents' outcomes computed by well-designed game rules (Zhang et al., 2024; DiGiovanni & Zell, 2021). Self-play has recently been explored to improve the capabilities of language models with no dependence on sophisticated human annotation. For instance, Cheng et al. (2024a) applied self-play of an adversarial language game to enhance LLMs' reasoning abilities, whereas the method only employs offline RL updates and remains confined to a simple word-based gaming environment. Other works, such as Fang et al. (2025) and Liang et al. (2025), generate problems from seed data but only train the solver models, leaving the problem-generation capability untrained and unsuitable for a co-evolutionary dynamic. More advanced and recent studies (Chen et al., 2025b; Huang et al., 2025a; Kuba et al., 2025) concurrently train both the proposer and solver models, demonstrating the effectiveness of this approach on diverse tasks like mathematics, code generation, and instruction following. However, these methods are limited by the LLMs' internal knowledge and not applicable under agentic scenarios. Our work makes two distinctions from prior self-play approaches. First, our proposer generates problems with accurate ground truth through an externally retrieved and verified validation pipeline. Second, with search tools, we equips the problem-proposer with external information, thereby breaking the limitations of the internal knowledge of LLMs.

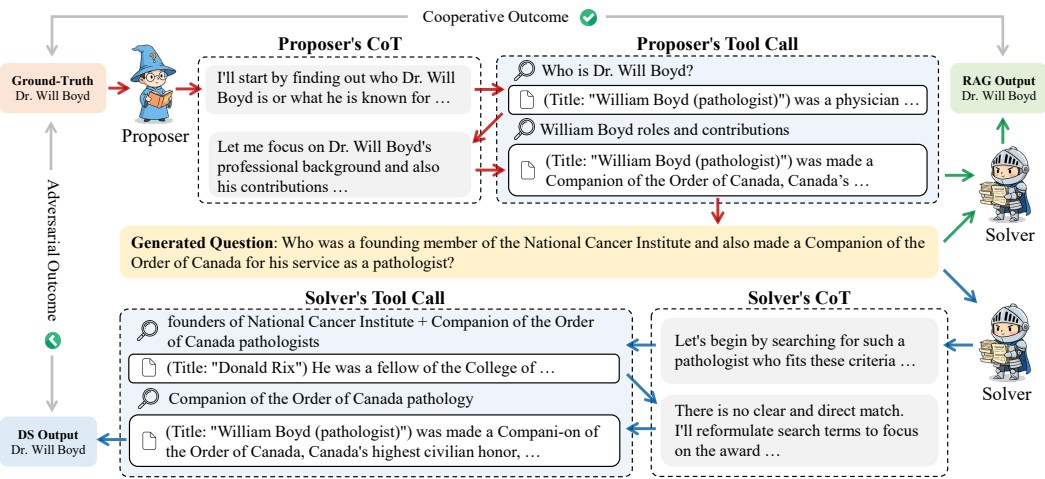

Figure 2: Examples of Search Self-play with a given ground-truth *"Dr. Will Boyd"*. Provided with the ground-truth, the *proposer* iteratively uses search tools to excavate implicit factual evidence, then generates a challenging search question. Then the solver leverages all search results from the proposer's trajectory as the RAG materials to predict the answer without searching, to validate the question's correctness. Once verified, the solver follows the ordinary deep search pipeline to explore the solution via multi-turn agentic rollout.

## 3 METHODOLOGY

Denote $\mathcal{N} := \{\boldsymbol{x} = (x^1, x^2, \ldots, x^N) : N \in \mathbb{N}_+, x^i \in \mathcal{V}, i = 1, 2, \ldots, N\}$ as the natural language sequence space, where $\mathcal{V}$ is the vocabulary set of tokens. An auto-regressive next-token prediction policy of LLM $\pi_\theta$ iteratively outputs next-token $x^{i+1} \sim \pi_\theta(\cdot|\boldsymbol{x}^{1:i})$ with $\theta$ as the model parameters, where $\boldsymbol{x}^{1:i} = (x^1, x^2, \ldots, x^i)$ is the length-$i$ prefix of the natual language sequence $\boldsymbol{x}$. A search agent trajectory can be written as: $\boldsymbol{\tau} = (\boldsymbol{x}, \boldsymbol{y}_1, \boldsymbol{o}_1, \boldsymbol{y}_2, \boldsymbol{o}_2, \ldots, \boldsymbol{y}_{T-1}, \boldsymbol{o}_{T-1}, \boldsymbol{y}_T)$, where $\boldsymbol{x} \in \mathcal{N}$ is the input prompt, each $\boldsymbol{y}_t \in \mathcal{N}$ is the LLM output at the $t$-th step, and $\boldsymbol{o}_t \in \mathcal{N}$ is the corresponding observation returned by the search tools at the $i$-th step. We model the search agent exploration as a token-level Markov decision process (Littman, 1994) $(\mathcal{S}, \mathcal{A}, \mathcal{T}, r)$. The state space $\mathcal{S}$ is naturally the language sequence space $\mathcal{N}$. The action space $\mathcal{A}$ is equivalent to the vocabulary set $\mathcal{V}$ for token-level action generation. The transition $\mathcal{T}$ directly appends the newly-generated token $y_t^{i+1}$ to the end of $\boldsymbol{y}_t^{1:i}$ if $\boldsymbol{y}_t^{1:i}$ has not formed a complete search tool call, or additionally appends the $t$-step observation $\boldsymbol{o}_t$ if $\boldsymbol{y}_t$ is finished. The reward function $r(\boldsymbol{\tau})$ assigns the outcome to trajectory $\boldsymbol{\tau}$ as the judgment of the agent's performance, where the design details are discussed in Section 3.2. Given a search agent system prompt $\boldsymbol{x}_{\text{sys}}$ and a user query $\boldsymbol{q}$, we can use a LLM policy $\pi_\theta$ to induce the search agent policy $u(\cdot|\boldsymbol{q}) = \pi_\theta(\cdot|\boldsymbol{x}_{\text{sys}}, \boldsymbol{q})$. For notation simplification, we denote $\boldsymbol{\tau} \sim u(\cdot|\boldsymbol{q})$ as collecting the trajectory $\boldsymbol{\tau}$ from the search agent policy $u(\cdot|\boldsymbol{q})$.

### 3.1 SEARCH SELF-PLAY DESIGN

We focus on exploring the benefits of self-play training for deep search agents, enabling LLMs to self-improve agent capabilities without additional supervision. To achieve this, we consider using the search agent to act as a *question proposer* to generate challenging questions via multi-turn deep search tool usage. Meanwhile, given the generated questions, we let the same LLM play as a *problem solver* to seek the answer, as ordinary deep search agents do. The proposer aims to generate increasingly challenging questions to puzzle the solver, whereas the solver is dedicated to improving its answer correctness, no matter how difficult the generated questions are. Based on the above rules, the search self-play can be regarded as a zero-sum adversarial game. We suppose both the proposer and the solver can evolve through this intense competition.

However, the above game rules can be easily hacked: the proposer can constantly generate incorrect questions so that the solver can never solve. Hence, to verify the correctness of the generated question from the proposer, we collect all the search results in the proposer's trajectory as the RAG documents, and let the solver answer without using search tools. If the proposer's question is correct

and the corresponding search actions are meaningful, with the RAG documents, the solver should already have sufficient information to correctly predict the answer. By this additional verification, we successfully avoid the search self-play game from hacking and degeneration. The verification constraint requires the proposer and the solver to cooperate, which enhances the SSP game with cooperation besides the proposing-solving competition. An example of the search self-play game is shown in Figure 2.

## 3.2 SEARCH SELF-PLAY MODELING

We use different system prompts $x_{\text{propose}}$ and $x_{\text{solve}}$ to let the LLM learning policy $\pi_\theta$ act as the proposer and the solver, respectively. Given a ground-truth answer $a$, the policy for the question proposer is $u(\cdot|a) = \pi_\theta(\cdot|x_{\text{propose}}, a)$. After the proposer generates a question $q$, the solver policy tries to settle the question with the policy $v(\cdot|q) = \pi_\theta(\cdot|x_{\text{solve}}, q)$. Denote $\tau$ and $\rho$ as the corresponding trajectories of the proposer and the solver, respectively. Then the adversarial self-play training objective is:

$$\min_u \max_v \mathbb{E}_{a^* \sim \mathcal{D}, \tau \sim u(\cdot|a=a^*), \rho \sim v(\cdot|q=\mathcal{Q}(\tau))}[r(\mathcal{A}(\rho), a^*)], \tag{1}$$

where $a^*$ is a ground-truth answer drawn from a pre-defined answer set $\mathcal{D}$. $\mathcal{Q}(\cdot)$ and $\mathcal{A}(\cdot)$ extract the generated question and predicted answer from the proposer trajectory $\tau$ and the solver trajectory $\rho$, respectively. $r(\mathcal{A}(\tau), a^*)$ is a binary outcome judgment function to check whether the solver's prediction $\mathcal{A}(\tau)$ and the ground-truth answer $a^*$ are semantically equivalent (which means $r(\mathcal{A}(\tau), a^*) = 1$). To ensure accurate judgment, we implement $r(\mathcal{A}(\tau), a^*)$ with an LLM-as-a-judge function, whose prompts and judge critics are described in Appendix E.

To make sure the generated question $q = \mathcal{Q}(\tau)$ is solvable and correct with respect to the ground-truth $a^*$, we need additional constraints. Therefore, we use the solver agent to $v(\cdot|q, O_T) = \pi_\theta(\cdot|x_{\text{solve}}, q, O_T)$ to verify the correctness of the generated question $\mathcal{Q}(\tau)$, where $O_T = (o_1, o_2, \ldots, o_T) = \mathcal{O}(\tau)$ is the collection of all the search results from the proposer trajectory. Then the proposer and the solver need to cooperate to maximize the solver's answer accuracy under RAG setups:

$$\max_u \mathbb{E}_{a^* \sim \mathcal{D}, \tau \sim u(\cdot|a=a^*), \sigma \sim v(\cdot|q=\mathcal{Q}(\tau), O_T=\mathcal{O}(\tau))}[r(\mathcal{A}(\sigma), a^*)]. \tag{2}$$

In practice, we find that jointly optimizing both the cooperation and competition objectives suffers from training inefficiency. Because the cooperative objection in equation 2 requires the proposed question to be completely correct, otherwise the optimization of equation 1 could lose its effectiveness due to the reward hacking. Therefore, we leverage rejection sampling (Liu et al., 2025a) for the cooperative objective instead. More specifically, we dynamically filter the generated questions with $r(\mathcal{A}(\sigma), a^*) = 1$ to collect a full batch of valid questions to optimize the adversarial objective in equation 1. Therefore, the overall training objective of search self-play is:

$$\min_u \max_v \mathbb{E}_{a^* \sim \mathcal{D}, \tau \sim u(\cdot|a=a^*), \rho \sim v(\cdot|q=\mathcal{Q}(\tau))}[r(\mathcal{A}(\rho), a^*)], \tag{3}$$

$$\text{s.t. } \mathbb{E}_{\sigma \sim v(\cdot|q=\mathcal{Q}(\tau), O_T=\mathcal{O}(\tau))}[r(\mathcal{A}(\sigma), a^*)] = 1.$$

## 3.3 SEARCH SELF-PLAY IMPLEMENTATION

We describe the details of the SSP optimization as equation 3 in Algorithm 1. As discussed in Section 3.2, invalid questions generated by the proposer will hinder the training effectiveness of SSP. Therefore, we applied two filtering strategies to improve the quality of generated questions: *rule-based filtering* and *RAG verification*.

Rule-based filtering ensures LLM has legal output for the question generation task. More specifically, each proposer's output should have a correct format for extracting the question (within `<question></question>` tags). Furthermore, we conduct several additional rule-based checks to pre-filter the low-quality questions and reduce the computational consumption before the RAG verification process. The bad cases to filter include: (1) empty question string; (2) no search tool invoked; (3) excessively short question; (4) containing the original answer in the question.

After the rule-based filtering, we applied the RAG verification process as described in Section 3.2. We collect the search results from the proposer's trajectory as the RAG documents, then let the

---

**Algorithm 1** Search Self-play training process.

---

**Require:** LLM policy $\pi_\theta$; ground-truth answer set $\mathcal{D}$; proposer and solver prompts $(\boldsymbol{x}_{\text{propose}}, \boldsymbol{x}_{\text{solve}})$.

1: **for** each parameter-updating step **do**
2:     Sample a batch of ground-truth answers $\{\boldsymbol{a}_i^*\}_{i=1}^B \sim \mathcal{D}$ with batch size $B$.
3:     Proposer generates candidate questions $\mathbb{Q} = \{\mathcal{Q}(\boldsymbol{\tau}_i)\}_{i=1}^B$ with each $\boldsymbol{\tau}_i \sim \pi_\theta(\cdot | \boldsymbol{x}_{\text{propose}}, \boldsymbol{a}_i^*)$.
4:     Filter out valid questions as $\mathbb{Q}^*$ with format rules and the RAG constraint:

$$r(\mathcal{A}(\boldsymbol{\sigma}_i), \boldsymbol{a}_i^*) = 1, \text{ for } \boldsymbol{\sigma}_i \sim \pi_\theta(\boldsymbol{x}_{\text{solve}}, \mathcal{Q}(\boldsymbol{\tau}_i)).$$

5:     **for** each question $\boldsymbol{q}_i \in \mathbb{Q}^*$ **do**
6:         Solver explores $n$ trajectories for solution: $\boldsymbol{\rho}_i^j \sim \pi_\theta(\cdot | \boldsymbol{x}_{\text{solve}}, \boldsymbol{q}_i), j = 1, 2, \ldots, n$
7:         Compute solver's reward of each trajectory: $r_{\text{solve},i}^j = r(\mathcal{A}(\boldsymbol{\rho}_i^j), \boldsymbol{a}_i^*)$
8:         Compute proposer's reward in expectation: $\bar{r}_{\text{propose},i} = 1 - \frac{1}{n} \sum_{j=1}^n r_{\text{solve},i}^j$
9:     **end for**
10:    Update $\pi_\theta$ with solver's trajectories and outcomes $\{(\boldsymbol{\rho}_i^j, r_{\text{solve},i}^j)\}$ via GRPO.
11:    Update $\pi_\theta$ with proposer's trajectories and outcomes $\{(\boldsymbol{\tau}_i, \bar{r}_{\text{propose},i})\}$ via REINFORCE.
12: **end for**

---

solver answer the generated question with the provided RAG materials. To further increase the robustness of the verification judgment, we mix some unrelated documents from other trajectories within the same batch to simulate more real RAG scenarios. Details and ablation studies about adding irrelevant RAG noises are discussed in Section 4.4.

When the outcome reward calculation finished, each generated question $\boldsymbol{q}_i$ had been tried by the solver for $n$ times, yielding a group of trajectories $\{\boldsymbol{\rho}_i^j\}_{j=1}^n$ and corresponding binary rewards $\{r_{\text{solve},i}^j\}_{j=1}^n$, where $r_{\text{solve},i}^j = r(\mathcal{A}(\boldsymbol{\rho}_i^j), \boldsymbol{a}_i^*)$. A natural updating method for the solver's policy $v$ is Group Relative Policy Optimization (GRPO) (Shao et al., 2024), which uses the average reward of the group as a baseline to reduce variance. The solver aims to maximize its reward, so its loss function for a given question $\boldsymbol{q}_i$ is:

$$\nabla_\theta \mathcal{L}_{\text{GRPO}}(\theta) = \frac{1}{B} \sum_{i=1}^B \left[ \frac{1}{n} \sum_{j=1}^n \frac{1}{|\boldsymbol{\rho}_i^j|} \sum_{t=1}^{|\boldsymbol{\rho}_i^j|} \nabla_\theta \log \pi_\theta(\rho_i^{j,t} | \boldsymbol{q}_i, \boldsymbol{\rho}_i^{j,1:t-1}) \cdot \hat{A}_i^j - \beta \nabla_\theta \text{KL}[\pi_\theta || \pi_{\text{ref}}] \right] \quad (4)$$

where the advantage $\hat{A}_i^j = r_{\text{solve},i}^j - \frac{1}{n} \sum_{k=1}^n r_{\text{solve},i}^k$ is calculated for the $j$-th trajectory of question $\boldsymbol{q}_i$.

Conversely, the proposer is updated to generate questions that are more challenging for the solver, which aligns with the min-max objective in equation 3. As defined in Algorithm 1, the proposer receives a high reward if the solver fails. We use the REINFORCE (Williams, 1992) algorithm to update the proposer's policy $u$. The loss function aims to increase the log-probability of generating trajectories that result in high proposer reward (i.e., low solver success rate):

$$\nabla_\theta \mathcal{L}_{\text{REINFORCE}}(\theta) = \frac{1}{B} \sum_{u=1}^B \left[ R(\boldsymbol{\tau}_i) \sum_{t=1}^{|\boldsymbol{\tau}_i|} \nabla_\theta \log \pi_\theta(\tau_i^t | \boldsymbol{a}_i^*, \boldsymbol{\tau}_i^{1:t-1}) \right], \quad (5)$$

where $R(\boldsymbol{\tau}_i) = 1 - \frac{1}{n} \sum_{j=1}^n r_{\text{solve},i}^j$. This update encourages the proposer to generate increasingly difficult questions to continuously challenge the task solver. Unlike prior question proposing methods, which only use LLMs' internal knowledge, our SSP utilizes interactions with external environments to acquire information for question generation. Moreover, our SSP verifies the correctness of the generated question with a verifiable RAG pipeline, which is more credible than previous synthetic methods, such as majority vote (Huang et al., 2025a).

## 4 EXPERIMENTS

### 4.1 EXPERIMENTAL SETUPS

**Benchmarks.** We evaluate SSP on seven widely-used question-answering benchmarks: NQ (Kwiatkowski et al., 2019), TriviaQA (Joshi et al., 2017), PopQA (Mallen et al., 2022), HotpotQA (Yang et al., 2018), 2WikiMultiHopQA (Ho et al., 2020), Musique (Trivedi et al., 2022), and Bamboogle (Press et al., 2022). Following the practice in prior works (Sun et al., 2025b;a; Zhao et al., 2025b; Gao et al., 2025a; Deng et al., 2025b; Tan et al., 2025), we randomly sample 500 question-answer (QA) pairs on each benchmark to reduce the evaluation overhead while maintaining statistical reliability. For Bamboogle (Press et al., 2022), all 125 test samples are used for evaluation.

**Baselines.** We select open-source pretrained LLMs of different sources and model sizes used for deep search, including Qwen2.5 (Yang et al., 2024a), LLaMA3.1 (Dubey et al., 2024), Qwen3 (Yang et al., 2025), Search-R1 (Jin et al., 2025b;a), ZeroSearch (Sun et al., 2025a), and R-Search (Zhao et al., 2025b).

**Search Tools.** A local E5 (Wang et al., 2022) retriever with a Wiki-2018 corpus (Karpukhin et al., 2020) is incorporated in our training and evaluation, which retrieves the top-3 related documents for each query. We limit search tool calls to 10 rounds for each trajectory.

**Evaluation Metrics.** Following recent work (Gao et al., 2025a), we adopt LLM-as-a-judge as standard metric for evaluation. Qwen2.5-32B-Instruct (Yang et al., 2024a) is deployed as the judge model. All results are reported in terms of pass@1 accuracy.

**Training Details.** We implemented our method using the SGLang asynchronous multi-turn tool-integrated rollout in VeRL (Sheng et al., 2024). The proposer is optimized using REIN-FORCE (Williams, 1992), while the solver is updated with GRPO (Shao et al., 2024). The learning rate is 1e-6 with 5 warmup steps. The global batch size and the mini-batch is 256 and 128, respectively. The maximum prompt length is 4,096 tokens, and the response length is set to 8,192 tokens. Each training horizon is within a range of 150 to 200 steps. The answer set $\mathcal{D}$ is sampled exclusively from public training sets (Jin et al., 2025b; Dong et al., 2025), detailed in Appendix A.4. More implementation details and prompts for both proposer and solver agents are provided in Appendix A & E. Full configurations are summarized in Table 4.

### 4.2 MAIN RESULTS

The main experimental results are summarized in Table 1. Across all settings, SSP consistently outperforms baseline counterparts on question-answering benchmarks. These results demonstrate, through consistent and substantial performance gains across a variety of models, training paradigms, and scales, that Search Self-play is a highly effective and versatile method for enhancing LLM agent capabilities.

Our primary finding is that SSP yields substantial improvements when training models from scratch without any external supervision. The gains are particularly pronounced for base models that have not undergone instruction tuning; for instance, applying SSP to Qwen2.5-7B-Base results in an impressive average improvement of 26.4 points, including a remarkable **+40.4** gain on TriviaQA. SSP also benefits instruction-tuned models, improving Qwen2.5-7B-Instruct by 8.0 points on average. Moreover, SSP proves to be model-agnostic, consistently enhancing models from different architectural families, including LLaMA-3.1 and Qwen3.

Additionally, SSP serves as an effective continual training strategy. Although strong open-source models have already been extensively trained on search-oriented tasks (e.g., Search-R1, R-Search), our method uniformly yields further performance improvements. Furthermore, the performance gain holds when scaling to larger models. When applying SSP to Qwen2.5-32B-Instruct, it achieves state-of-the-art results on five of the seven benchmarks. These competitive results demonstrate the consistent effectiveness of SSP for training agents across diverse model sizes, architectures, and initial agentic performances.

Table 1: Main experimental results. SSP delivers strong gains across from-scratch training, generalization across architectures, continual training on search-specialized agents, and scaling to larger models. All scores are on a 100-point scale. Bold (black) indicates the better score within each baseline v.s. +SSP pair.

| Method | GeneralQA | | | Multi-HopQA | | | | Avg |
|---|---|---|---|---|---|---|---|---|
| | NQ | TriviaQA | PopQA | HotpotQA | 2Wiki | MuSiQue | Bamboogle | |
| **From-Scratch Training on Base and Instruct Models** | | | | | | | | |
| Qwen2.5-7B-Base | 32.0 | 33.2 | 25.0 | 18.0 | 10.8 | 11.0 | 26.4 | 22.3 |
| + SSP | $\mathbf{54.2}^{+22.2}$ | $\mathbf{73.6}^{+40.4}$ | $\mathbf{56.0}^{+31.0}$ | $\mathbf{52.8}^{+34.8}$ | $\mathbf{33.2}^{+22.4}$ | $\mathbf{24.0}^{+13.0}$ | $\mathbf{47.2}^{+20.8}$ | $\mathbf{48.7}^{+26.4}$ |
| Qwen2.5-7B-Instruct | 44.2 | 64.0 | 36.4 | 45.0 | 32.8 | 16.8 | 51.2 | 41.5 |
| + SSP | $\mathbf{54.8}^{+10.6}$ | $\mathbf{73.4}^{+9.4}$ | $\mathbf{51.8}^{+15.4}$ | $\mathbf{51.8}^{+6.8}$ | $\mathbf{38.8}^{+6.0}$ | $\mathbf{21.2}^{+4.4}$ | $\mathbf{54.4}^{+3.2}$ | $\mathbf{49.5}^{+8.0}$ |
| **Generalization Across Model Families** | | | | | | | | |
| LLaMA-3.1-8B | 50.2 | 65.2 | 45.8 | 34.6 | 19.4 | 11.4 | 30.4 | 36.7 |
| + SSP | $\mathbf{58.0}^{+7.8}$ | $\mathbf{75.8}^{+10.6}$ | $\mathbf{55.4}^{+9.6}$ | $\mathbf{44.2}^{+9.6}$ | $\mathbf{34.4}^{+15.0}$ | $\mathbf{16.2}^{+4.8}$ | $\mathbf{40.0}^{+9.6}$ | $\mathbf{46.3}^{+9.6}$ |
| Qwen3-8B | 53.6 | 76.0 | 50.8 | 54.2 | 48.0 | 26.6 | 58.4 | 52.5 |
| + SSP | $\mathbf{56.0}^{+2.4}$ | $\mathbf{78.2}^{+2.2}$ | $\mathbf{55.0}^{+4.2}$ | $\mathbf{58.0}^{+3.8}$ | $\mathbf{51.5}^{+3.5}$ | $\mathbf{28.0}^{+1.4}$ | $\mathbf{67.2}^{+8.8}$ | $\mathbf{56.3}^{+3.8}$ |
| **Continual Training on Search-Specialized Agents** | | | | | | | | |
| ZeroSearch-7B | 52.2 | 66.6 | 50.2 | 43.2 | 34.6 | 17.6 | 40.8 | 43.6 |
| + SSP | $\mathbf{54.2}^{+2.0}$ | $\mathbf{69.0}^{+2.4}$ | $\mathbf{53.0}^{+2.8}$ | $\mathbf{44.0}^{+0.8}$ | $\mathbf{37.2}^{+2.6}$ | $\mathbf{19.6}^{+2.0}$ | $\mathbf{44.0}^{+3.2}$ | $\mathbf{45.9}^{+2.3}$ |
| Search-R1-7B | 56.6 | 75.4 | 57.2 | 58.2 | 45.2 | 29.6 | 55.2 | 53.9 |
| + SSP | $\mathbf{57.8}^{+1.2}$ | $\mathbf{78.0}^{+2.6}$ | $\mathbf{58.4}^{+1.2}$ | $\mathbf{60.4}^{+2.2}$ | $\mathbf{45.6}^{+0.4}$ | $\mathbf{30.6}^{+1.0}$ | $\mathbf{59.2}^{+4.0}$ | $\mathbf{55.7}^{+1.8}$ |
| R-Search-7B | 50.8 | 71.0 | 53.8 | 54.0 | 56.4 | 29.8 | 53.6 | 52.8 |
| + SSP | $\mathbf{52.4}^{+1.6}$ | $\mathbf{74.2}^{+3.2}$ | $\mathbf{56.8}^{+3.0}$ | $\mathbf{54.2}^{+0.2}$ | $\mathbf{58.0}^{+1.6}$ | $\mathbf{31.4}^{+1.6}$ | $\mathbf{55.2}^{+1.6}$ | $\mathbf{54.6}^{+1.8}$ |
| **Scaling to Larger Models** | | | | | | | | |
| Qwen2.5-14B-Instruct | 56.0 | 77.0 | 53.8 | 57.0 | 48.4 | 26.6 | 64.8 | 54.8 |
| + SSP | $\mathbf{57.4}^{+1.4}$ | $\mathbf{77.8}^{+0.8}$ | $\mathbf{54.6}^{+0.8}$ | $\mathbf{61.2}^{+4.2}$ | $\mathbf{49.4}^{+1.0}$ | $\mathbf{28.0}^{+1.4}$ | $\mathbf{69.6}^{+4.8}$ | $\mathbf{56.9}^{+2.1}$ |
| Qwen2.5-32B-Instruct | 58.0 | 78.4 | 53.4 | 57.0 | 48.4 | 27.4 | 63.2 | 55.1 |
| + SSP | $\mathbf{62.6}^{+4.6}$ | $\mathbf{82.8}^{+4.4}$ | $\mathbf{55.0}^{+1.6}$ | $\mathbf{62.8}^{+5.8}$ | $\mathbf{49.2}^{+0.8}$ | $\mathbf{32.0}^{+4.6}$ | $\mathbf{69.6}^{+6.4}$ | $\mathbf{58.5}^{+3.4}$ |

## 4.3 Self-play versus Fixed-Opponent Training

The co-evolution of the proposer and solver is critical for pushing the frontier of agent capability. We investigate this core hypothesis through an ablation study comparing the complete Search Self-play framework against two fixed-opponent schemes: training only the solver, denoted as *Solver-Only*, and training only the proposer, denoted as *Proposer-Only*. As shown in Table 2, the results clearly demonstrate the superiority of SSP. Our SSP achieves the highest average score, substantially outperforming both fixed-opponent variants. The training dynamics, analyzed in Figure 3, reveal the reasons behind this performance gap.

**Solver-Only:** figure 3 (a) reveals the underlying issue with Solver-Only. The solver's in-game reward rapidly saturates near 0.9, indicating that it quickly masters the static distribution of tasks from the fixed proposer. In the lack of a progressively challenging curriculum, the solver begins to overfit. This is confirmed by its performance on held-out evaluation sets (Figure 3(b) and (c)), where scores on NQ and 2Wiki initially increase but then decrease over time.

**Proposer-Only:** conversely, the Proposer-Only setting shows different limitations. While its in-game reward also rises, its evaluation performance on NQ and 2Wiki initially declines before a slight recovery. We attribute this partial recovery to the proposer learning general tool-use skills, which incidentally aid the fixed solver. This effect is more pronounced on simpler GeneralQA benchmarks like NQ, where this setting eventually surpasses Solver-Only. However, this general skill enhancement is insufficient for complex multi-hop reasoning, resulting in lower performance on Multi-HopQA datasets compared to the Solver-Only setup.

In contrast to the flawed dynamics of fixed-opponent training, our complete SSP framework facilitates a stable co-evolution. As shown in Figure 3(a), the solver's in-game reward initially rises, but unlike the saturating curve of the Solver-Only setting, it later experiences a slight decline. This dip is not a sign of performance degradation, but rather crucial evidence of the proposer's co-evolution: it has learned to generate more difficult tasks that challenge the improving solver, thus reducing its success rate. This dynamic creates a robust and adaptive curriculum where task difficulty perpetually adjusts to the solver's current agentic level, preventing overfitting and forcing continuous

Table 2: Ablation on training schemes. The complete search self-play (**+SSP**) significantly outperforms fixed-opponent variants, underscoring the necessity of co-evolution for robust performance gains.

| Method | GeneralQA | | | Multi-HopQA | | | | Avg. |
|---|---|---|---|---|---|---|---|---|
| | NQ | TriviaQA | PopQA | HotpotQA | 2Wiki | MuSiQue | Bamboogle | |
| Qwen2.5-7B-Instruct | 44.2 | 64.0 | 36.4 | 45.0 | 32.8 | 16.8 | 51.2 | 41.5 |
| +SSP (Solver-Only) | 45.2 | 68.2 | 46.6 | 47.2 | 32.6 | 21.0 | 48.8 | 44.2 |
| +SSP (Proposer-Only) | 52.4 | 69.0 | 50.4 | 44.8 | 28.8 | 14.2 | 32.0 | 41.7 |
| **+SSP** | **54.8** | **73.4** | **51.8** | **51.8** | **38.8** | **21.2** | **54.4** | **49.5** |

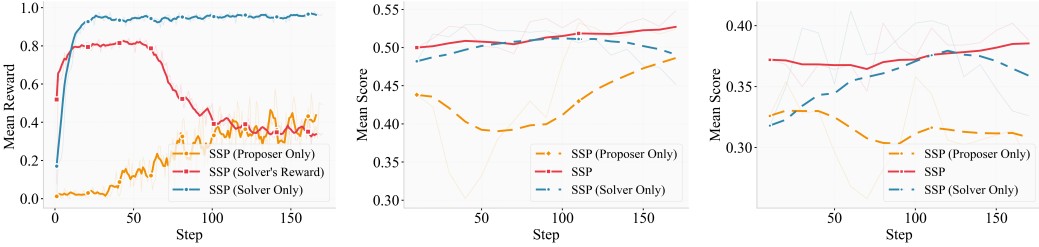

(a) In-Game Training Reward     (b) Evaluation Score on NQ     (c) Evaluation Score on 2Wiki

Figure 3: Training dynamics of different SSP variants. (a) shows the in-game reward. (b) and (c) display the evaluation accuracy on the held-out NQ and 2Wiki datasets over training steps.

learning. Consequently, this internal adversarial pressure reflects on the stable and sustained performance gains of benchmarks in Figure 3 (b) and (c). This confirms that the mutual evolution between the proposer and solver is decisively superior to fixed-opponent training.

## 4.4 ABLATION ON RAG VERIFICATION

Another key component of our SSP framework is the RAG verification, which validates each proposed question is correct and answerable given all evidences collected by the proposer. We conduct an ablation study to quantify the impact of the RAG verification and optimize its configuration at the same time. All ablation studies are conducted on Qwen2.5-7B-Instruct, with results demonstrated in Table 3.

First, we compare our SSP against a variant with no RAG verification. As shown in the results, removing RAG verification leads to a significant performance decay, particularly on GeneralQA benchmarks. This result confirms our hypothesis that the RAG verification is crucial for question quality controlling, which effectively prunes invalid questions and prevents the solver from being trained on noisy or incorrect data.

Next, we examine the impact of injecting noisy documents into the search materials for RAG verification. These documents are randomly sampled from other trajectories within the same training batch and are *not* retrieved by the proposer during its search trajectory. The goal of this strategy is to prevent the proposer from hacking the self-play game: without irrelevant materials, the proposer can generate easy-for-RAG but hard-for-deep-search questions based on a fixed range of augmented documents. For instance, if the proposer explored 5 biographical portraits, a hacking question generation could be *"Who is the earliest-born individual?"*, which is easy for RAG to answer

Table 3: Ablation on the RAG verification. Performance (Avg. Score) improves with a moderate number of noisy context documents.

| Config. | GeneralQA | Multi-HopQA |
|---|---|---|
| No RAG Verifi. | 49.5 | 36.7 |
| *RAG verification w/ Noisy Docs:* | | |
| 0 Noisy Docs | 58.5 | 38.2 |
| 1 Noisy Docs | 58.5 | 36.9 |
| **4 Noisy Docs** | **60.0** | **41.6** |
| 7 Noisy Docs | 57.8 | 35.9 |

within the fixed 5 portraits, but with insufficient conditions for deep search agents to solve with unrestricted search freedom. Additional analyses of hacking question examples are provided in Appendix F. Injecting noisy documents into the RAG verification context poses a greater challenge for the solver, which forces the solver to validate the ground truth answer in the presence of both

relevant and irrelevant information. This increased difficulty discourages the proposer from generating ambiguous questions. Besides, unknown to the randomly injected documents, the proposer is forced to produce more robust questions whose answers are strongly and uniquely supported by the searched evidence, rather than ones that are trivially verifiable in a clean, noise-free context. Based on the experimental results in Table 3, adding four noisy documents yields the best performance across benchmarks. In contrast, exhaustive noise injection (e.g., 7 random documents) degrades accuracy, due to increased confusion during verification. We therefore choose the number of noise documents to 4 in our main experiments.

## 5 CONCLUSION

We introduce Search Self-play (SSP), a self-evolving reinforcement learning approach for deep search agents, in which the LLM policy acts as both a question proposer and a problem solver. In the proposed game, both agents utilize multi-turn search engine interactions, where the proposer aims to generate more difficult questions with verifiable ground-truths and the solver tries to predict the answer accurately. The correctness of generated questions is verified by the cooperation between the proposer and the solver via a retrieval-augmented generation (RAG). With the well-designed competition and cooperation, both the proposer and solver significantly improve themselves in the SSP games. Extensive experiments demonstrate that SSP consistently enhances search agent performance across diverse benchmarks under both from-scratch and continuous training setups, without requiring any external human supervision. These results highlight the potential of self-play as a scalable and data-efficient paradigm for agentic LLM training, paving the path for more efficient and self-sustaining RL methods in complicated agentic application scenarios.

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

# A   IMPLEMENTATION DETAILS

## A.1   TRAINING HYPERPARAMETER

The hyperparameters for our main experiments are detailed in Table 4. Parameters not explicitly mentioned in the table adhere to the default settings provided by the veRL framework (Sheng et al., 2024).

Table 4: Experimental hyperparameter configuration. The table is divided into two sections: Base Settings for the main training process, and specific settings for Search Self-play.

| Base Settings | |
|---|---|
| **Parameter** | **Value** |
| Global Batch Size | 256 |
| Mini-batch Size | 128 |
| Learning Rate | 1e-6 |
| Learning Rate Warmup Steps | 5 |
| Max Prompt Length | 4096 |
| Max Response Length | 8192 |
| KL Loss Coefficient | 0.01 |
| Validation Temperature | 0.0 |
| Rollout Temperature | 1.0 |
| **Search Self-play Settings** | |
| **Parameter** | **Value** |
| *— Proposer —* | |
| Proposer Warm-up Steps | -1 (disabled) |
| Proposer Advantage Estimator | REINFORCE |
| Proposer Samples (n) | 1 |
| *— Solver —* | |
| Solver Advantage Estimator | GRPO |
| Solver Samples (n) | 5 |
| *— Other —* | |
| Batch Sampling Strategy | Replay Buffer (Periodic Reset) |
| Use RAG Verification | True |
| Noisy RAG Documents | 4 |

## A.2   REWARDS DESIGN

**Solver.** A simple binary outcome reward is designed for Solver:

$$r_{\text{solve}} = \mathbf{1}(\boldsymbol{a} = \boldsymbol{a}^*) \tag{6}$$

where $\boldsymbol{a}^*$ is the ground-truth answer and $\mathbf{1}(\cdot)$ is the indicator function that checks for equality between the predicted and true answers.

**Proposer.** The solver attempts to answer the same question $n$ times, where the average success rate for that question is $\bar{r}_{\text{solve}} = \frac{1}{n}\sum_{i=1}^{n} r_{\text{solve}}^i$. Thus, the reward for the proposer can be formulated as:

$$r_{\text{propose}} = 1 - \bar{r}_{\text{solve}} \tag{7}$$

Tool-integrated rollout is an interactive sequence of reasoning and tool invocation, with the search tool providing external information. Following Search-R1 (Jin et al., 2025b;a), we mask the content within the `<information></information>` tags to exclude their loss computation during training to maintain stability. The responses of both the proposer and solver are required to strictly follow the target format, using tags such as `<think>`, `<search>`, `<answer>`, and `<question>`. Responses that deviate from the format will receive no reward.

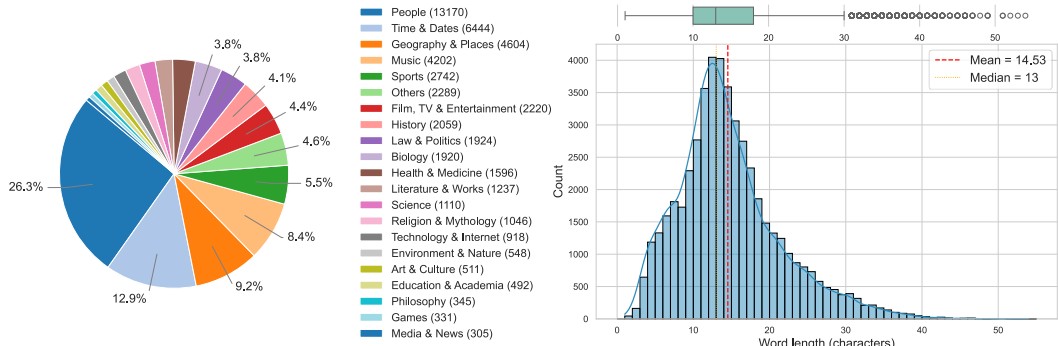

Figure 4: Diversity and length distribution of pre-defined answer set $\mathcal{D}$.

## A.3 BASELINES

We use the following baseline models for continuous RL training: Search-R1 (Jin et al., 2025b;a), ZeroSearch (Sun et al., 2025a), and R-Search (Zhao et al., 2025b). We start the SSP training from the best-performing checkpoint as reported in their respective papers. For example, Search-R1-7B corresponds to checkpoint `SearchR1-nq_hotpotqa_train-qwen2p5-7b-em-ppo-v0p2`, ZeroSearch-7B to checkpoint `ZeroSearch_wiki_V2_Qwen2.5_7B`, and R-Search-7B to checkpoint `R-Search-7b-grpo`.

During the question generation process, the number of valid questions in each batch is significantly reduced after applying the filtering strategies. To mitigate this, we replenish the batch using dynamic sampling (Yu et al., 2025), which ensures that the rewards within a training batch are less sparse, thus contributing to a more stable training process. To enhance sampling efficiency in experiments, the default RAG solver in RAG verification is Qwen2.5-32B-Instruct.

## A.4 CONSTRUCTION OF PRE-DEFINED ANSWER SET $\mathcal{D}$

The entire set $\mathcal{D}$ is sampled exclusively from public training data. Specifically, we construct $\mathcal{D}$ by randomly sampling ground-truth answers from (1) the Search-R1 training set (i.e., the training sets of NQ and HotpotQA), and (2) the ground-truth answers used in ARPO's released training set (Dong et al., 2025). We sample 50,000 ground-truth answers to form the final answer set $\mathcal{D}$. The word length distribution is presented in Figure 4, showing an averaged length of 14.53. $\mathcal{D}$ spans a broad range of topics, including People (26.3%), Time & Dates (12.9%), Geography & Places (9.2%), Music (8.4%), Sports (5.5%), Film/TV/Entertainment (4.4%), History (4.1%), Law & Politics (3.8%), Biology (3.8%), and others.

## B ADDITIONAL EXPERIMENTAL RESULTS

### B.1 ABLATION ON BATCH SAMPLING STRATEGIES

In our SSP framework, the proposer's generation process is stochastic, and not all generated questions pass the online filter. This can result in training batches smaller than the target batch size, leading to sparse reward signals and potential training instability. To address this, we investigate four distinct batch sampling strategies below to ensure a full batch is always available for the RL update step. All experiments are conducted on the Qwen2.5-7B-Base model, with results summarized in Table 5.

- **Dummy Padding:** Invalid slots in a batch are filled with a generic, non-informative "dummy" problem. This simple approach provides no learning signal for the padded slots.

- **Dynamic Resampling:** The proposer continues to generate new questions until a full batch of valid problems is collected. It ensures every sample in the batch is novel, but it is computationally expensive when the valid question pass rate is low.

Table 5: Ablation study on batch sampling strategies. Performance is evaluated on Qwen2.5-7B-Base. The Periodic Reset strategy yields the best results, highlighting the importance of striking a balance between data reuse and novelty.

| Method | NQ | TriviaQA | PopQA | HotpotQA | 2Wiki | MuSiQue | Bamboogle | Avg. |
|---|---|---|---|---|---|---|---|---|
| Qwen2.5-7B-Base (Baseline) | 32.0 | 33.2 | 25.0 | 18.0 | 10.8 | 11.0 | 26.4 | 22.3 |
| Dummy Padding | 48.4 | 68.8 | 49.6 | 40.8 | 22.6 | 19.0 | 40.8 | 41.4 |
| Dynamic Resampling | 48.4 | 66.4 | 45.8 | 44.6 | 31.4 | 17.6 | 42.4 | 42.4 |
| Replay Buffer (Full Reuse) | 51.2 | 67.8 | 49.0 | 46.0 | 31.0 | 17.8 | **48.0** | 44.4 |
| **Replay Buffer (Periodic Reset)** | **54.2** | **73.6** | **56.0** | **52.8** | **33.2** | **24.0** | 47.2 | **48.7** |

- **Replay Buffer (Full Reuse):** We maintain a replay buffer of all previously generated valid questions. Invalid slots are filled by sampling from the buffer, which guarantees a dense training signal but risks solver overfitting and proposer policy stagnation.

- **Replay Buffer (Periodic Reset):** This strategy is used to reproduce our main experimental results. It is identical to *Replay Buffer (Full Reuse)*, but the replay buffer is cleared every 10 training steps, balancing the efficiency of reuse with the need for data novelty.

As shown in Table 5, the choice of strategy has a profound impact on training outcomes. The *Dummy Padding* approach yields the smallest improvement over the baseline. Its low performance can be attributed to severe reward sparsity. With many invalid proposals, both the proposer and solver receive fewer learning signals per iteration, hindering effective optimization. *Dynamic Resampling* performs slightly better, as it guarantees a full batch of novel, valid questions. However, it comes at a high, often prohibitive, computational cost, as it requires repeated generation cycles.

The *Replay Buffer (Full Reuse)* strategy provides a significant performance boost, improving the average score from 42.4 to 44.4, which allows the solver to learn more thoroughly from each valid question generated by the proposer by reusing it for multiple training updates. It verifies the reward signal and enhances training efficiency. However, its gains are ultimately limited, for unbounded reuse allows the solver to train on the same questions too many times, leading to overfitting on the static pool of questions within the ever-growing buffer. Concurrently, the proposer's learning signal diminishes as the solver masters these old questions, potentially causing policy degradation.

The *Replay Buffer (Periodic Reset)* strategy emerges as the clear winner, achieving the highest scores across nearly all benchmarks and boosting the average score to 48.7, which represents an effective trade-off between sufficient data exposure and novelty. Reusing questions for a limited period allows the solver to learn sufficiently from each generated task, ensuring the reward signal remains dense. However, periodically clearing the buffer prevents the solver from learning the same questions too many times, thus mitigating the overfitting observed with full reuse. Concurrently, this forces the proposer to continuously generate novel questions to populate the fresh buffer, maintaining a strong co-evolutionary pressure. The result validates the effectiveness of *Replay Buffer (Periodic Reset)*, as it fosters the most stable and effective self-play training.

## B.2 TRAINING DYNAMICS OF SOLVER

We provide a granular view into the training process by analyzing the training dynamics of our SSP framework on the Qwen2.5-7B-Base model, configured with the *Replay Buffer (Periodic Reset)* strategy. Figure 5 illustrates core metrics in the self-play training, demonstrating how the agent's behavior and performance co-evolve.

As shown in Figure 5a, the average number of search tool calls per trajectory steadily increases over time, indicating that through search self-play, the agent learns to conduct more extensive and complex multi-step searches to solve problems, significantly enhancing its tool-use capabilities. Simultaneously, Figure 5b shows that the solver's response length also grows during the training, suggesting it learns to generate more detailed and comprehensive answers. In contrast, the prompt length remains relatively stable, indicating consistent task/question generation of the proposer.

Figures 5c and 5d demonstrate a consistent and significant improvement in accuracy across both GeneralQA and Multi-HopQA datasets as training progresses. Notably, the slope of performance improvement gradually decreases in later training stages. This plateau is partially attributable to a

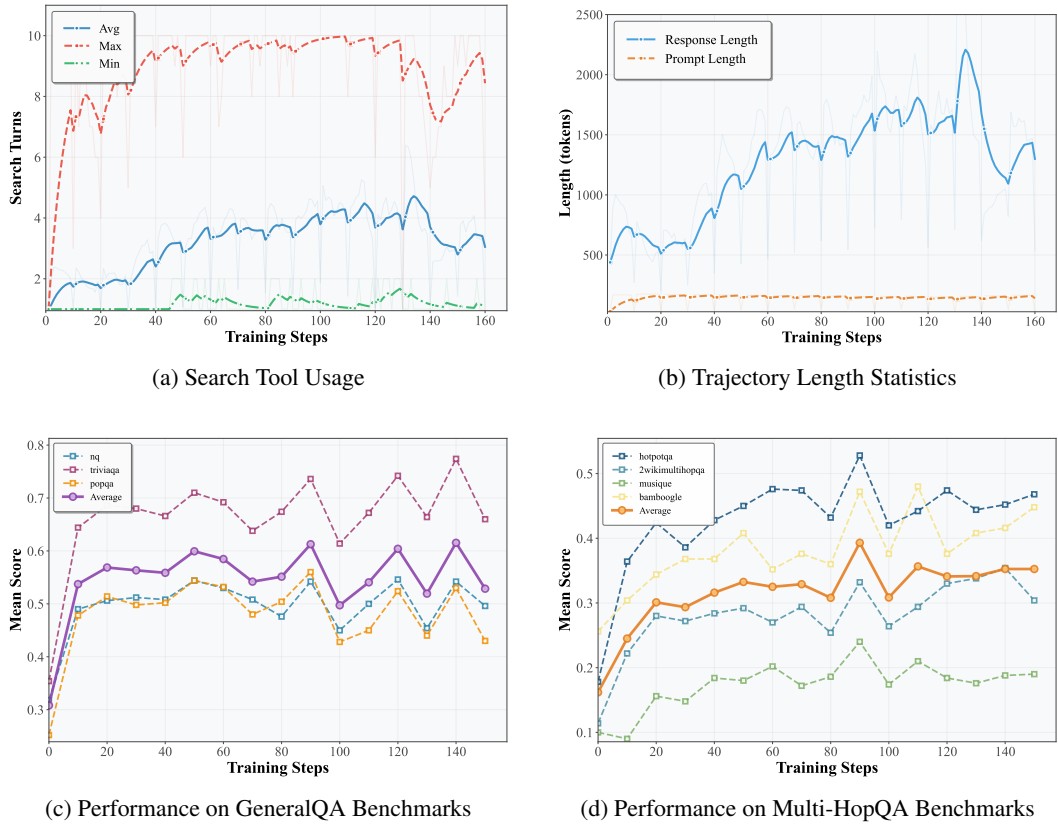

(a) Search Tool Usage

(b) Trajectory Length Statistics

(c) Performance on GeneralQA Benchmarks

(d) Performance on Multi-HopQA Benchmarks

Figure 5: Training dynamics of SSP with Qwen2.5-7B-Base. (a) The agent learns to use the search tool more frequently. (b) Solver response length increases while prompt length remains stable. (c, d) Evaluation scores on both GeneralQA and Multi-HopQA datasets show continuous improvement during training.

resource-imposed constraint: the maximum number of search steps was capped at 10 to conserve computational resources, preventing the agent from exploring even deeper reasoning paths. We believe that scaling the search step constraint could unlock further performance improvements.

## B.3 TRAINING DYNAMICS OF PROPOSER

To gain deeper insights into the proposer's behavior during SSP training, we conduct a comprehensive analysis of its evolution across multiple dimensions. Figure 6 presents four key aspects of the proposer's development throughout the training process.

The analysis reveals several key insights into the proposer's learning dynamics. First, as shown in Figure 6a, the proposer demonstrates a consistent upward trend in search tool utilization throughout training. The progressive increase in search frequency indicates that the proposer develops more sophisticated information-gathering strategies, enabling it to construct increasingly complex question scenarios that require deeper exploration of external knowledge sources. Additionally, Figure 6b demonstrates a steady improvement in question validation success rate, rising from near 0% to approximately 50% by the end of training. This upward trend indicates that the proposer learns to generate higher-quality questions that successfully pass verification, reflecting improved question formulation skills and better alignment with the verification constraints. In designing the SSP framework, we have prioritized high precision in the verification process, even at the cost of some recall. While this means a subset of valid questions might be filtered out, it effectively ensures that only high-confidence samples enter the self-play loop. This selective mechanism helps maintain a clean and stable training environment, which we find essential for facilitating robust co-evolution between the proposer and solver.

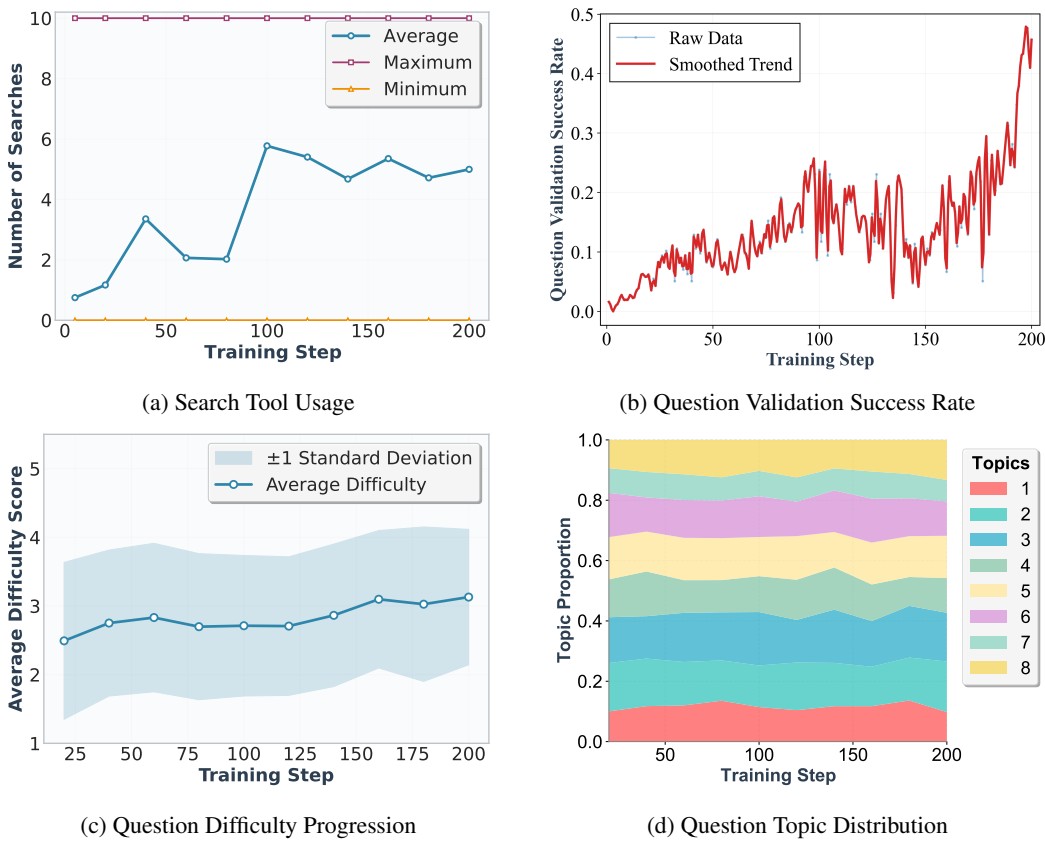

Figure 6: Comprehensive analysis of proposer dynamics during SSP training. (a) The proposer gradually increases its search tool usage, demonstrating enhanced exploration capabilities. (b) Question validation success rate shows steady improvement, indicating enhanced question quality over training. (c) Generated question difficulty progressively increases, evidencing adaptive curriculum learning. (d) Topic distribution remains well-balanced across training, indicating broad question coverage.

Figure 6c reveals a clear progression in the difficulty of generated questions throughout training. To systematically observe the evolution of question difficulty, we employ DeepSeek-V3.2 (Liu et al., 2024)to evaluate and score the generated questions using a structured difficulty assessment prompt (detailed in Appendix E). The proposer exhibits an adaptive curriculum learning behavior, gradually increasing the complexity and challenge level of its generated questions. This progressive difficulty scaling demonstrates the proposer's ability to provide increasingly sophisticated challenges to the solver, facilitating the co-evolutionary dynamics that drive mutual improvement in the SSP framework. The upward trend in question difficulty, coupled with maintained topic diversity, confirms that the proposer successfully balances challenge progression with comprehensive coverage.

To assess the diversity and coverage of generated questions, we employ Latent Dirichlet Allocation (LDA) (Blei et al., 2003) clustering to analyze the topical distribution of the questions generated by proposer. As demonstrated in Figure 6d, the topic distribution remains remarkably balanced across different training phases. The consistent proportional representation across multiple topic clusters indicates that the proposer successfully maintains broad domain coverage and avoids bias toward specific question types, thereby ensuring comprehensive curriculum development.

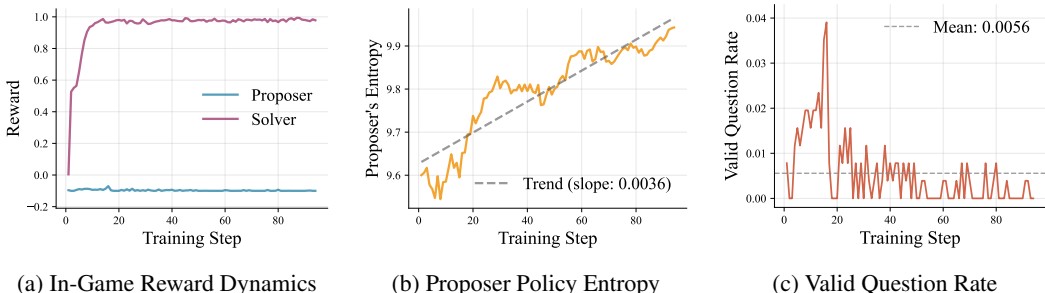

(a) In-Game Reward Dynamics  (b) Proposer Policy Entropy  (c) Valid Question Rate

Figure 7: Training dynamics of SSP when the proposer receives a negative reward for format errors. (a) The proposer's reward (blue) trends to -0.1 as it stops producing valid questions, while the solver's reward (purple) spuriously increases due to overfitting on a question pool. (b) The proposer's policy entropy steadily rises as the agent tries to escape the negative rewards, leading to more random and less valid outputs. (c) Consequently, the rate of generating valid questions collapses, halting productive self-play training and demonstrating the instability caused by punitive rewards.

## B.4 PROPOSER'S REWARD DESIGN

We analyze the sensitivity of our Search Self-play framework to the reward function with an experiment on the effect of a punitive reward structure for the proposer. In our main configuration, the proposer receives a zero reward for generating an invalid or malformed question. Besides, we introduce a small penalty, setting the reward as -0.1 for any question that fails the online filter. The proposer is optimized using the REINFORCE algorithm with a single sample per prompt ($n = 1$), while the solver is updated using GRPO, and we adopt the *Replay Buffer (Full Reuse)* strategy for batch sampling.

The results, shown in Figure 7, demonstrate that the seemingly minor change leads to a catastrophic failure of the training process. The proposer's average reward becomes sparse and progressively declines (Figure 7a), directly corresponding to a collapse in the valid question generation rate, which plummets towards 0 (Figure 7c). This phenomenon can be explained as a negative feedback loop: the penalty for format errors encourages the agent to explore away from its current policy. However, this exploration, manifested as an increase in policy entropy (Figure 7b), makes the generation more random and thus more likely to produce invalid outputs. This "death spiral" effectively halts the creation of new training instances. Meanwhile, the solver's reward appears to increase, which is a misleading artifact of overfitting. As the supply of new, valid questions from the proposer dwindles, the solver is repeatedly trained on a small, static buffer of past questions, failing to generalize its capabilities. This experiment critically underscores that the proposer's reward design is paramount for stable co-evolution in SSP; a punitive approach can destabilize the entire self-play dynamic, highlighting the need for a carefully calibrated reward scheme.

## B.5 ABLATION STUDY ON RL ALGORITHMS

To investigate the impact of different reinforcement learning algorithms on our SSP framework, we conduct comprehensive experiments comparing various combinations of RL algorithms for the proposer and solver roles. The experimental results are presented in Table 6. We set the per-query rollout budget to n=5 whenever GRPO is used for either role (proposer or solver), and n=1 when a role is trained by REINFORCE (RF).

Training both roles with GRPO achieves the highest average accuracy (50.9), slightly outperforming our default RF–GRPO configuration (49.5), but with a substantial computational cost: the per-step generation time rises from 83.4 s to 504.4 s, *i.e.*, approximately 6× slower. The improvements are modest and appear across several datasets (e.g., +1.2 on NQ, +2.0 on PopQA, +3.0 on 2Wiki, +1.2 on MuSiQue, +2.4 on Bamboogle), with a small drop on TriviaQA (−0.6). Given the pronounced increase in runtime and resource consumption, the GRPO–GRPO configuration is the most expensive option despite only marginal gains over the default.

Table 6: Evaluation results and rollout generation time comparisons on different RL algorithms for proposers and solvers, where RF denotes REINFORCE.

| Proposer | Solver | NQ | TriviaQA | PopQA | HotpotQA | 2Wiki | MuSiQue | Bamboogle | Avg. | T/step(s) |
|----------|--------|------|----------|-------|----------|-------|---------|-----------|------|-----------|
| RF | GRPO | 54.8 | 73.4 | 51.8 | 51.8 | 38.8 | 21.2 | 54.4 | 49.5 | 83.4 |
| RF | RF | 53.0 | 71.2 | 53.6 | 46.2 | 28.0 | 13.2 | 28.8 | 42.0 | 9.1 |
| GRPO | RF | 51.2 | 69.2 | 49.8 | 47.6 | 37.0 | 21.6 | 48.8 | 46.5 | 50.1 |
| GRPO | GRPO | 56.0 | 72.8 | 53.8 | 52.4 | 41.8 | 22.4 | 56.8 | 50.9 | 504.4 |

Among the remaining settings (i.e., configurations other than GRPO–GRPO), the first configuration (RF–GRPO) delivers the best average performance (49.5) while maintaining reasonable generation time (83.4 s/step). Using GRPO for the proposer while training the solver with RF substantially degrades accuracy to 46.5, even though the per-step time is lower (50.1 s). Notably, the RF–RF pairing yields the lowest average performance (42.0), despite the shortest per-step time (9.1 s). These results suggest that placing GRPO on the solver side is more effective than on the proposer side: multi-trajectory credit assignment during solving directly benefits solution quality, whereas stronger exploration in proposing alone does not yield improvements unless the solver can reliably evaluate and solve the proposed queries.

**Takeaways.** The GRPO–GRPO pair is slightly more accurate than the default RF–GRPO but is by far the slowest and most resource-intensive configuration. For practical training budgets, the small accuracy gains do not justify the $\sim 6\times$ increase in generation time. Among all other configurations, RF–GRPO achieves the best trade-off between effectiveness and efficiency and remains our default choice.

## C  USE OF LARGE LANGUAGE MODELS

Large language models were used to aid in the writing and polishing of this manuscript. Specifically, LLMs assisted with language refinement and improving clarity. All core content and intellectual contributions are solely the work of the authors.

## D  SELF-PLAY EXAMPLES

Table 7, 8, 9, 10, 11, and 12 provide examples of trajectories in our SSP game.

Table 7: Self-play example 1: question proposing.

**Proposer** creates a question that leads to the answer *Roger (alien)*

`<think>`Given that I have a single search requirement, it's essential to design a straightforward but intricate-enough route to come to the term "Roger (alien)" that would normally require a direct search. This will differentiate "Roger (alien)" from other similar or general terms to ensure the question is precise and unambiguous.

Starting with "Roger (alien)," this seems like a reference to a specific character or entity. "Roger" is a common name. The term "alien" specifically refers to an alien from other galaxies or universes.

I'll brainstorm some 'associated concepts' related to "Roger (alien)." If it is a specific cultural reference or a name from specific media content, I can build my questions around these concepts.

Let's begin by searching for 'associated' or 'preceding' facts which connect to "Roger (alien)" without specifying the answer directly.`</think>`

`<search>` Roger alien `</search>`

`<information>` (Title: "Roger (American Dad!)") Roger is a fictional character in the adult animated sitcom ""American Dad!"", voiced by Seth MacFarlane. The character was created and designed by Seth MacFarlane. Roger is a centuries-year-old grey space alien living with the Smith family, around whom the show revolves. Having . . .

(Title: "Big O (mecha)") locks itself and lets Roger out because R. Dorothy is in danger. In ""Enemy Is . . .

(Title: "Roger (Hellboy)") Roger the Homunculus is a fictional character featured in the . . . `</information>`

`<think>`It appears that the most prominent "Roger (alien)" refers to a character named Roger from the animated sitcom, "American Dad!," who is known to be an extraterrestrial being from another planet. Given this, the associated concept is now clear, and we can build the question accordingly. We need to find a particular instance or unique detail linked to Roger from American Dad! which would require a solver to search to reach the final answer.`</think>`

`<search>` Roger alien american dad! `</search>`

`<information>` (Title: "Roger (American Dad!)") Roger is a fictional character in the adult animated . . .

(Title: "Roger (American Dad!)") be shown on the series. Although in the episode ""Lost in Space"", a brief clip . . .

(Title: "American Dad!") alien race. To date, Roger's birth planet has yet to be revealed . . . `</information>`

`<think>`Based on the information gathered, I see that there's a notable character detail that can be used—the fact that Roger has a backstory linked to Roswell crash. Given that it's quite specific and links directly to the search results, we can use this idea to form our single-search question.`</think>`

`<question>` Who is the grey alien character from American Dad! that was involved in the Roswell incident in 1947? `</question>`

Table 8: Self-play example 1: question answering.

**Solver** answers the question raised by Proposer: *Who is the grey alien character from American Dad! that was involved in the Roswell incident in 1947?*

`<think>`To answer this question, I need to identify a grey alien character from American Dad! that has a connection to the Roswell incident in 1947. I'm not sure off-hand which character is known for this, as the show features many alien characters. Searching for the details will likely retrieve the relevant information.`</think>`

`<search>` Grey alien character from American Dad! Roswell incident 1947 `</search>`

`<information>` (Title: "Grey alien") as ""visitors"", appear in two episodes of ""South Park"", and Roger . . .

(Title: "Grey alien") small-bodied beings with smooth grey-colored skin, enlarged hairless heads and large . . .

(Title: "Roger (American Dad!)") Roger is a fictional character in the adult animated sitcom ""American Dad!"", voiced by Seth MacFarlane. The character was created and designed by Seth MacFarlane. Roger is a centuries-year-old grey space alien living with the Smith family, around whom the show revolves. Having lived on Earth since crashing in Roswell, New Mexico in 1947, Roger came to live with the Smiths after rescuing main character Stan Smith at Area 51 four years prior to the beginning of the series. . . . `</information>`

`<think>`Based on the provided information, the grey alien character from American Dad! that was involved in the Roswell incident in 1947 is Roger. The text mentioned that Roger is a grey space alien living with the Smith family, and he has been on Earth since crashing in Roswell, New Mexico in 1947.`</think>`

`<answer>` Roger `</answer>`

Table 9: Self-play example 2: question proposing.

**Proposer** creates a question that leads to the answer *Castle Drogo*

`<think>`The target answer is Castle Drogo. To create a question that requires three sequential searches, I need to find an associated concept for the final answer and reverse-engineer the question from there.

First, let's identify an associated concept for Castle Drogo. Castle Drogo is a 20th-century castle located in the South West of England, situated in Devon. The associated concept could be the architect behind it, Sir Edwin Lutyens.

Next, I'll search for information connecting Sir Edwin Lutyens to Castle Drogo to find a 'preceding fact' for the third search.`</think>`

`<search>` Sir Edwin Lutyens and Castle Drogo `</search>`

`<information>` (Title: "Castle Drogo") Castle Drogo Castle Drogo is a country house and castle near Drewsteignton, Devon, England. Constructed between 1911 and 1930, it was the last castle to be built in England. The client was Julius Drewe, the hugely successful founder of the Home and Colonial Stores. Drewe chose the site in the belief that it formed part of the lands of his supposed medieval ancestor, Drogo de Teigne. The architect he chose to realise his dream was Edwin Lutyens, then at the height of his career. Lutyens lamented Drewe's determination to . . .

(Title: "Castle Drogo") purely decorative. Additionally, the castle had electricity and lifts from the outset, with power being supplied by two turbines on the river below. The castle is a Grade I listed building. . . .

(Title: "Castle Drogo") the village of Drewsteignton in the belief . . . `</information>`

`<think>`From the search results, we know that Edwin Lutyens designed Castle Drogo. The preceding fact for the third search could be the architect who designed Castle Drogo.

Now, I need to find another piece of information that leads logically to Edwin Lutyens as the answer, which will be the second search. I know that Sir Edwin Lutyens was a notable 20th-century English architect, known for his numerous significant works, including collaborations with Gertrude Jekyll. Let's explore his collaboration with Jekyll further.

Lutyens frequently collaborated with the famous garden designer Gertrude Jekyll. I'll find a specific project or fact related to this collaboration.`</think>`

`<search>` Edwin Lutyens and Gertrude Jekyll `</search>`

`<information>` (Title: "Gertrude Jekyll") wife Julia Hammersley. Her younger brother, Walter Jekyll (an Anglican priest; sometime Minor Canon of Worcester Cathedral and Chaplain of Malta) . . .

(Title: "Gertrude Jekyll") was awarded the Victoria Medal of Honour of the Royal Horticultural Society in 1897 and the Veitch Memorial Medal of the society in 1929. Also in 1929, she was given the George Robert White Medal . . .

(Title: "Hestercombe House") the house by Henry Hall in the 1870s. The Edwardian garden was laid out by Gertrude Jekyll and Edwin Lutyens between 1904 and 1906 for . . . `</information>`

`<think>`From the search results, I can see that Edwin Lutyens and Gertrude Jekyll collaborated on several projects, particularly Hestercombe House. The preceding fact for the second search could be a well-known collaboration project of Edwin Lutyens and Gertrude Jekyll.

For the final step, I need to identify an associated concept related to Hestercombe House that leads to Edwin Lutyens as the answer. I can create a sequential search question that starts from a well-known architectural or garden feature associated with Hestercombe House.

The Hestercombe Garden is a well-known aspect. The next fact for the first search could be related to a designer or architect's masterpiece associated with Hestercombe Garden.`</think>`

`<search>` Hestercombe Garden architect `</search>`

`<information>` (Title: "Hestercombe House") petrol and air. When the house and gardens were inherited . . .

(Title: "Hestercombe House") the house by Henry Hall in the 1870s. The Edwardian garden was laid out by Gertrude Jekyll and Edwin Lutyens between 1904 and 1906 for the Hon E.W.B. Portman, resulting in a garden ""remarkable for the bold, concise pattern of its layout, and for the minute attention to detail everywhere to be seen in the variety . . .

(Title: "Hestercombe House") the hardy herbaceous planting espoused by Miss Jekyll. Lutyens also designed the orangery about 50 m east of the main house between 1904–09, which is now Grade I listed . . . `</information>`

`<think>`The search results indicate that Hestercombe House and Garden were designed by Edwin Lutyens and Gertrude Jekyll. This association can lead us to the architect Edwin Lutyens in a search starting from a prominent aspect of the Hestercombe Garden. The question can start by referencing a key feature of Hestercombe Garden and logically lead to Edwin Lutyens and later Castle Drogo.

To formulate the question, let's start referring to the specific feature, such as the orangery built by Edwin Lutyens.`</think>`

`<question>` What is the name of the 20th-century castle in Devon, England, designed by the architect who, alongside Gertrude Jekyll, created the famous Edwardian garden at Hestercombe House, known for its listed orangery? `</question>`

Table 10: Self-play example 2: question answering.

---

**Solver** answers the question raised by Proposer: *What is the name of the 20th-century castle in Devon, England, designed by the architect who, alongside Gertrude Jekyll, created the famous Edwardian garden at Hestercombe House, known for its listed orangery?*

`<think>`The question is asking for the name of a 20th-century castle in Devon, England. This castle is designed by an archite ct who also worked with Gertrude Jekyll on the Hestercombe House gardens. The Hestercombe House gardens are f amous for their Edwardian style and especially known for their listed orangery.

Given this information, I need to find:

1. The architect who collaborated with Gertrude Jekyll on the Hestercombe House gardens.

2. A 20th-century castle in Devon, England, designed by that architect.`</think>`

`<search>` architect who collaborated with Gertrude Jekyll on Hestercombe House gardens `</search>`

`<information>` (Title: "Hestercombe House") the house by Henry Hall in the 1870s. The Edwardian garden was laid out by Gertrude Jekyll and Edwin Lutyens between 1904 and 1906 for the Hon E.W.B. Portman, resulting in a garden ""remarkable for the bold, concise pattern of its layout, and for the minute attention to detail everywhere to be seen in the variety and imaginative handling of contrasting materials, whether cobble, tile, flint, or thinly coursed local stone"". Jekyll and Lutyens were leading participants of the Arts and Crafts movement. Jekyll is remembered for her outstanding designs and subtle, painterly approach to the arrangement of the gardens . . .

(Title: "Hascombe Court") Hascombe Court Hascombe Court is a estate in Hascombe, Surrey, best known for its vast garden designed by Gertr ude Jekyll. Hascombe Court is listed Grade II on the National Heritage List for England, and its gardens are also Grad e II listed on the Register of Historic Parks and Gardens. . . .

(Title: "Gertrude Jekyll") over 300 photographs taken by Jekyll. From 1881, when she laid out the gardens for Munstead House, built for her mother by John James Stevenson, Jekyll provided designs or planned planting for some four hundred gardens. More than half were directly commissioned, but many were . . . `</information>`

`<think>`Doc 1 mentions that Edwin Lutyens and Gertrude Jekyll collaborated on the garden at Hestercombe House. With this in mind, I can now search for a 20th-century castle in Devon, England designed by Edwin Lutyens.`</think>`

`<search>` 20th-century castle in Devon, England designed by Edwin Lutyens `</search>`

`<information>` (Title: "Castle Drogo") Castle Drogo Castle Drogo is a country house and castle near Drewsteign-ton, Devon, England. Constructed between 1911 and 1930, it was the last castle to be built in England. The client was Julius Drewe, the hugely successful founder of the Home and Colonial Stores. Drewe chose the site in the belief that it formed part of the lands of his supposed medieval ancestor, Drogo de Teigne. The architect he chose to realise his dream was Edwin Lutyens, then at the height of his career. Lutyens lamented Drewe's determination to . . .

(Title: "Devon County War Memorial") The Devon County War Memorial is a First World War memorial, designed by Sir Edwin Lutyens and situated on Cathedral Green in Exeter, the county town of Devon, in the south west of England. It is one of fifteen War Crosses designed by Lutyens with similar characteristics . . .

(Title: "Devon County War Memorial") memorials were built across Britain. Amongst the most prominent designers of memorials was the architect Sir Edwin Lutyens, described by Historic England as ""the leading English architect of his generation"". Prior to the First World War, Lutyens established . . . `</information>`

`<answer>` Castle Drogo `</answer>`

---

Table 11: Self-play example 3: question proposing.

---

**Proposer** creates a question that leads to the answer *George Smith*

<think>To create a question that leads to the answer "George Smith," which requires three searches, I need to work backwards and establish a logical sequence of information that will lead to this name. Let's start by identifying an associated concept or related information.

1. **Associated Concept**: George Smith might be a person involved in a specific historical event, a notable figure in a certain field, or someone connected to a particular organization or place. For this, I'll explore who George Smith might be and find a notable fact or place relevant to this person.

2. **Search Process**:
- **Search #3**: Start with the target "George Smith." I need to identify an event or place associated with him.
- **Search #2**: Once the preceding fact is discovered from Search #3, I'll use that information to identify another intermediary fact.
- **Search #1**: Finally, will determine the starting point using the information from Search #2.

Let's start with Search #3 to establish the initial associated concept.</think>

<search> Who is George Smith and what is he known for? </search>

<information> (Title: "George P. Smith (politician)") George Peter Smith (August 12, 1873 – November 29, . . .

(Title: "George Smith (historian)") George Smith (31 August 1800 – 30 August 1868) was an English businessman . . .

(Title: "George Smith (artist)") George Smith (1713/14 – 7 September 1776) was . . . </information>

<think>From the search results, George Smith (politician) from Alberta provides a specific context. George P. Smith played a role in the development of Duhamel, Alberta, and the establishment of influential relationships with indigenous peoples. I will use this in the backward search chain.

**Search #3**: Identify a fact related to George P. Smith.</think>

<search> What significant role did George P. Smith play in the community of Duhamel, Alberta? </search>

<information> (Title: "George P. Smith (politician)") George Peter Smith (August 12, 1873 – November 29, . . .

(Title: "George P. Smith (politician)") with fraud as they believed that the money which was . . .

(Title: "George Wilbert Smith") George Wilbert Smith (April 24, 1855 – August 1, 1931) . . . </information>

<think>From the information, George P. Smith played a significant role in founding the Trading Store in Duhamel, Alberta, and developing it into a hub for fur trading with indigenous peoples.

**Search #3 Key Fact**: George P. Smith bought and established a Trading Store in Duhamel, Alberta.

Next, we will now build on this fact in Search #2.</think>

<search> What role did trading stores play in the development of settlements like Duhamel, Alberta, in the early 20th century? </search>

<information> (Title: "Duhamel, Alberta") Duhamel is a hamlet in the . . .

(Title: "Duhamel, Alberta") soil, and vegetation within the circles had apparently been . . .

(Title: "Exchange District") was a new route for shipping goods from Eastern Canada . . . </information>

<think>From the search, trading stores played a crucial role in connecting settlers with resources and indigenous peoples. Tr ading stores in early settlements were central to economic and social development.

**Search #2 Key Fact**: Trading stores were central hubs for economic activities, distributing goods and establishing trade relations with indigenous peoples.

Finally, we build the starting point for Search #1.</think>

<search> Which early 20th-century Alberta town was a key trading center due to the presence of a significant fur-trading activity? </search>

<information> (Title: "History of Alberta") providing an important channel for the flow . . .

(Title: "Lamson & Hubbard Trading Company") Lamson & Hubbard Trading Company (also referred to as . . .

(Title: "Northern Traders Company") The Northern Traders Company was an enterprise . . . </information>

<think>From the search, Alberta towns like Red Deer could have been key trading centers due to fur trading activity. Red Deer's role fits well because it was an economic hub with connections to the indigenous communities.

**Search #1 Key Fact**: Red Deer, Alberta, was a key trading center due to its fur-trading activity.

Now I can construct the final question ensuring it meets all the established requirements and necessitates precisely three searches to reach the final answer.</think>

<question> What was the name of the politician who established a crucial trading store in the early 20th century, which transformed a small community into an important trading hub for fur and developed strong relationships with the indigenous peoples in Alberta? </question>

---

Table 12: Self-play example 3: question answering

**Solver** answers the question raised by Proposer: *What was the name of the politician who established a crucial trading store in the early 20th century, which trans formed a small community into an important trading hub for fur and developed strong relationships with the indigenous peoples in Alberta?*

`<think>`The question asks for a politician who established a trading store in Alberta, Canada in the early 20th century. This politician seems to have had substantial influence in the region, not just in politics but also in trade, and built strong relationships with indigenous peoples. The region mentioned, Alberta, was an important area for fur trading during this period.

Given that the politician founded a crucial trading store, it's likely that they were known for their contributions to Canadian history, particularly in the province of Alberta. Notable figures in Alberta during the early 20th century would be relevant here, particularly those linked with economic development through trade or industry, and who engaged in dialogue or relationships with indigenous groups.

To identify this specific politician, more precise historical information about fur trading in Alberta in the early 20th ce ntury is required, especially someone known for such a significant act of setting up a trading hub.`</think>`

`<search>` politician who established a trading store in Alberta early 20th century `</search>`

`<information>` (Title: "James Ramsey (politician)") James ""Merchant Prince"" Ramsey (April 4, 1864 – December 23, 1939) was a provincial politician and business man from Alberta, Canada . . .

(Title: "Alexander Livingstone (Alberta politician)") Alexander Livingstone (August 2, 1884 – October 26, 1944) was a Canadian politician and municipal councillor in Edmonton, Alberta. Livingstone was born in Bathgate, Scotland . . .

(Title: "Cornelius Hiebert") Cornelius ""Don"" Hiebert (August 2, 1862 – March 21, 1919) was a politician from Alberta, Canada. Hiebert was born in Sea of Azov, Russia and lived in Russia . . . `</information>`

`<think>`The initial search results did not directly identify a politician who established a fur trading store or created a significa nt trading hub in early 20th-century Alberta. James Ramsey and Alexander Livingstone were notable business and political figures, but they are primarily mentioned for their work in department stores in Edmonton, not for a specific trading store related to fur or indigenous relations. Cornelius Hiebert was involved in setting up a general store in a founding phase of a village, but not detailed circumstances specifically in trading of fur.

A second search is needed with a more concentrated focus on fur traders and key individuals who had influence in relation to indigenous communities.`</think>`

`<search>` politician from Alberta early 20th century established fur trading store `</search>`

`<information>` (Title: "James Ramsey (politician)") James ""Merchant Prince"" Ramsey (April 4, 1864 . . .

(Title: "Cornelius Hiebert") Cornelius ""Don"" Hiebert (August 2, 1862 – March 21, 1919) was . . .

(Title: "Alexander Livingstone (Alberta politician)") Alexander Livingstone (August 2 . . . `</information>`

`<think>`Despite the multiple searches, there appears a lack of direct information providing an exact match regarding a prominent fur trader politician in Alberta from the early 20th century. The information given points primarily to notable business and political figures linked with department stores or general stores in Alberta but not specifically to fur trading or close relations to indigenous communities as mentioned in the question. As the information is still scattered and does not directly answer the specific individual queried, a reconsideration and possibly with a broader search is required to uncover the correct specific person's name and connection with fur trading in Alberta.`</think>`

`<search>` politician who established a fur trading store in Alberta early 1900s `</search>`

`<information>` (Title: "George P. Smith (politician)") George Peter Smith (August 12, 1873 – November 29, 1942) was a politician and former Minister of the Crown from Alberta, Canada. He was born in Lieury, Ontario and came west in 1901 while working for an insurance company. After visiting Duhamel (a mission settlement . . .

(Title: "Cornelius Hiebert") Cornelius ""Don"" Hiebert (August 2, 1862 – March 21, 1919) was . . .

(Title: "Alexander Livingstone (Alberta politician)") Alexander Livingstone (August 2 . . . `</information>`

`<think>`The information provided initially presents George P. Smith as someone who bought a trading store in 1902 after visiting Duhamel in Alberta and developed a relationship with influential settlers and with Métis and Saulteaux peoples. He formed a lucrative fur trading business, leading to the establishment of a trading hub important for fur trade. This matches well with the criteria outlined in the question. The second document did not specifically mention fur trade, while the third one was about a prominent figure, but not specifically in the context of fur trading. So it strongly indicates that George P. Smith is the person sought.

Given the direct relevance of George P. Smith's background and activities in developing a fur trading hub and formin g relationships with indigenous communities in Alberta, the answer to the question appears to be George P. Smith.`</think>`

`<answer>` George P. Smith `</answer>`

# E   PROMPTS

Prompts used in our search self-play experiments are listed as follows.

---

**Proposer Prompt**

You are an expert question creator. Your primary task is to reverse-engineer a challenging question from a given answer. The question you create must require a solver to perform 'n' sequential searches to solve it. I will provide you with the target answer and the required number of searches, 'n'.

**Your Creation Process & Tools:**
**1. Analyze Scope and Target:** Begin by analyzing the provided 'Answer' (your target) and the required number of searches, 'n' (the path's length). This establishes your final destination and the complexity of the logical chain you need to construct.

**2. Build the Question by Working Backwards:**
   This is the core of the process. You will start from the destination and work your way back to the starting point, step by step.
   **2.1. The Crucial First Step: Connection and Discovery**
      Start with the final 'Answer', but do not search for the answer itself directly.
      Instead, first analyze the 'Answer'. Brainstorm and identify a closely related yet distinct 'Associated Concept' (e.g., a related historical event, a key figure, a geographic location, a unique attribute, its parent category, etc.).
      Perform an exploratory search with the goal of finding the 'bridging information' that connects this 'Associated Concept' to your final 'Answer'.
      From your search results, extract a unique, verifiable 'preceding fact'. This is a piece of information that, when searched, would logically lead a user to your final 'Answer'. This 'preceding fact' becomes the answer to Search #n-1.
   **2.2. Iterate Backwards**
      Now, treat this newly found 'preceding fact' as your new target.
      From this point on, you can search for this new target directly to find the preceding piece of information that leads to it. This becomes the answer to the next search in the backward chain (Search #n-2).
   **2.3. Construct the Full Chain**
      Continuously repeat the iterative process from Step 2.2, using each new fact as the target for the next backward search, until you have constructed a complete logical chain of 'n' links.
      The very first piece of information you uncover in this process (the one at the start of the chain) will become the initial clue for your question.

**3.** You must conduct reasoning inside `<think>` and `</think>` first every time you get new information. After reasoning, if you find you lack some knowledge, you can call a search engine by `<search>` query `</search>`, and it will return the top searched results between `<information>` and `</information>`. You can search as many times as you want. If you find no further external knowledge needed, you can directly provide the answer inside `<question>` and `</question>` without detailed illustrations. For example, `<question>` xxx `</question>`.

**Here are three example questions:**
Question 1: {example1}
Question 2: {example2}
Question 3: {example3}

**Critical Rules:**
**1. Strictly Fact-Based:** You must not create questions based on assumptions. The entire logical path to the solution must be grounded in the information you find through searching.

---

**2. No Spoilers:** The question must not contain any direct clues that reveal the answer or the intermediate steps.

**3. Search is Mandatory:** The question must be impossible to answer from general knowledge alone. It must necessitate the search process you have designed.

**4. Adhere to Search Count:** The number of searches required to solve the question must precisely match the specified 'search count'.

**5. Unique Answer:** The designed question must be deterministic, leading to a single, unambiguous final answer. The clues at each step must be precise enough to prevent a solver from reasonably arriving at a different, valid conclusion.

The answer I provided is: {answer}.

You need to create a question that requires {n} searches.

When you have enough information to construct a question, please first check whether the constructed question meets all requirements, especially whether the question is too simple. After checking that all conditions are met, you need to provide the final constructed question in your final response, placing the final question between `<question>` and `</question>` tags, for example: `<question>` ... `</question>`.

---

### RAG Solver Prompt

Answer the given question based on the provided materials. You should first conduct very concise reasoning within 50 words, and then directly provide your answer without detailed illustrations after saying 'Answer:'.

**Materials:** {materials}

**Question:** {question}

---

### Solver Prompt (Search-R1 / ZeroSearch / Qwen series / LLaMA 3.1)

You are a helpful and harmless assistant.

Answer the given question. You must conduct reasoning inside `<think>` and `</think>` first every time you get new information. After reasoning, if you find you lack some knowledge, you can call a search engine by `<search>` query `</search>` and it will return the top searched results between `<information>` and `</information>`. You can search as many times as your want. If you find no further external knowledge needed, you can directly provide the answer inside `<answer>` and `</answer>`, without detailed illustrations. For example, `<answer>` Beijing `</answer>`. Question: {question}

---

### Solver Prompt (R-Search)

You are a helpful assistant that can solve the given question step by step. For each step, start by explaining your thought process. If additional information is needed, provide a specific query enclosed in `<search>` and `</search>`. The system will return the top search results within `<observation>` and `</observation>`. You can perform multiple searches as needed. When you know the final answer, use `<original_evidence>` and `</original_evidence>` to provide all potentially relevant original information from the observations. Ensure the information is complete and preserves the original wording without modification. If no searches were conducted or observations were made, omit the evidence section. Finally, provide the final answer within `<answer>` and `</answer>` tags. {question}

---

**LLM-as-a-judge Prompt**

You are a professional judge who evaluates the correctness of answers based on given criteria.
Please determine whether the model's answer is consistent with the reference answer:

Question: {question}
Model Answer: {prediction}
Reference Answer: {ground_truth}

Evaluation Criteria:
1. The model answer must accurately respond to the question and be consistent with the reference answer in meaning.
2. For numerical questions, the values must be equal or very close.
3. For textual questions, the core meaning must be correct.
4. Differences in wording or language are allowed as long as the core answer is the same.
5. If the model answer includes the correct answer and does not contain conflicting information, it is also considered correct.

Please respond only with "Correct" or "Wrong". Do not provide any additional explanation.

---

**Question Difficulty Evaluator Prompt**

You are a search-problem difficulty evaluator. Your task: given a single search-type question, return a strict JSON containing only two fields:
"overall_difficulty": an integer from 1 to 5 (1 = easiest, 5 = hardest) "reasoning": a short explanation (1–2 sentences) describing why you gave that score, focusing on required reasoning and expected search effort
DIFFICULTY SCALE:

- 1: Very simple factual questions
- 2: Basic factual questions requiring single search
- 3: Questions requiring moderate search or basic reasoning
- 4: Complex questions requiring multiple searches or reasoning steps
- 5: Very complex multi-step questions requiring extensive research and reasoning

Examples:
Easy valid question: "What process causes the continents to drift apart?" Expected output: {{"overall_difficulty": 1, "reasoning": "Direct factual question requiring single-step search for a well-known geological concept."}}
Hard valid question: "In the 19th century, a work published by a French writer sparked judicial proceedings for 'corrupting public morals.' This work was subsequently adapted into multiple film versions. In the 1991 version, what flavor did the lead actress insist on using for the prop poison in the scene where her character dies by suicide?" Expected output: {{"overall_difficulty": 5, "reasoning": "Requires complex multi-step reasoning: identifying the French work and writer, finding the 1991 film adaptation, then locating very specific behind-the-scenes details about prop choices."}}
Here is the question to evaluate: Question: {question}
Output requirements — STRICT JSON only, no extra text or fields. Return only valid JSON.

---

## F    HACKING QUESTION CASES

The example in Table 13 illustrates a significant problem with the proposer's question formulation: non-uniqueness of answers. The question "Temptations singer" is inherently ambiguous because the Temptations, as a famous Motown group, had multiple singers throughout their history, including David Ruffin, Eddie Kendricks, Dennis Edwards, and many others. The question essentially asks for

"a singer from a band," which obviously has multiple valid answers. However, the RAG Solver has the documents that prominently feature one particular member (in this case, Otis Williams), making it appear as if there's a single correct answer. While a RAG-based solver can succeed due to the limited and biased retrieval results, a solver working independently without access to these specific documents would likely struggle to determine which of the many possible Temptations singers is the "correct" answer.

Table 13: Example of hacking questions: question with non-unique answers.

---

**Proposer** creates a question that leads to the answer *Otis Williams*

`<think>`I need to create a question that leads to "Otis Williams" as the answer.

Let's begin by searching for 'associated' or 'preceding' facts which connect to "Otis Williams".`</think>`

`<search>` otis williams `</search>`

`<information>` Doc 1 (Title: "Otis Williams") Otis Williams Otis Williams (born Otis Miles Jr.; October 30, 1941) is an American baritone singer. Nicknamed ""Big Daddy"", he is occasionally also a songwriter and a record producer. Williams is best known as the founder and last surviving original member of the Motown vocal group The Temptations, a group in which he continues to perform; he also owns the rights to the Temptations name. Williams was born Otis Miles, Jr. in Texarkana, Texas to Otis Miles and Hazel Louise Williams . . .

Doc 2 (Title: "Charles Malik Whitfield") Charles Malik Whitfield Charles Malik Whitfield (born August 1, 1972) is an American actor. Whitfield is best known for his performance as Otis Williams in the television miniseries, ""The Temptations"" (1998), for which he was nominated for a NAACP Image Award for Outstanding Actor in a Television Movie or Mini-Series . . .

Doc 3 (Title: "Johnny Otis") Johnny Otis Johnny Otis (born Ioannis Alexandres Veliotes; December 28, 1921 – January 17, 2012) was an American singer, musician, composer, arranger, bandleader, talent scout, disc jockey, record producer, television show host, artist, author, journalist, minister, and impresario . . . `</information>`

`<think>`It appears that the most prominent "Otis Williams" refers to a singer from The Temptations, a famous Motown group.`</think>`

`<question>` Temptations singer `</question>`

---

Table 14: Example of hacking questions: question with non-unique answers (wrong answer).

---

**Solver** answers the question raised by Proposer: *Temptations singer?*

`<think>`I need to find out which singer from the Temptations group. I'll search for it.`</think>`

`<search>` Temptations singer `</search>`

`<information>` Doc 1 (Title: "Terry Weeks") Terry Weeks Terry Wayne Weeks (born December 23, 1963) is an American R&B and soul singer who is currently one of the lead singers of the legendary Motown . . .

Doc 2 (Title: "Otis Williams") Otis Williams Otis Williams (born Otis Miles Jr.; October 30, 1941) is an American baritone singer. Nicknamed ""Big Daddy"", he is occasionally also a songwriter and a record producer. Williams . . .

Doc 3 (Title: "The Temptations") The Temptations The Temptations are an American vocal group who released a series of successful singles and albums with Motown Records during the 1960s and 1970s. Their work with producer Norman Whitfield, beginning with the Top 10 hit single ""Cloud Nine"" in October 1968, pioneered psychedelic soul, and was significant in the evolution of R&B and soul music. The band members are known for their choreography, distinct harmonies, and dress style. Having sold tens of millions of albums, the Temptations are among the most successful groups in popular music. Featuring five male vocalists and dancers (save for brief periods with . . . `</information>`

`<think>`I found out that The Temptations are a vocal group. Now I need to find out which singer from the Temptations.`</think>`

`<search>` who is the singer in The Temptations `</search>`

`<information>` Doc 1 (Title: "David Ruffin") David Ruffin David Eli Ruffin (born Davis Eli Ruffin, January 18, 1941 – June 1, 1991) was an American soul singer and musician most famous for his work as one of the lead singers of The Temptations (1964–68) during the group's ""Classic Five"" period as it was later known. He was the lead voice on such . . .

Doc 2 (Title: "The Temptations") The Temptations The Temptations are an American vocal group who released a series of successful singles and albums with Motown Records during the 1960s and 1970s. Their work with producer Norman Whitfield, beginning with the Top 10 hit single ""Cloud Nine"" in October 1968, pioneered psychedelic soul, and was significant in the evolution of R&B and soul music. The band members are known for their choreography, distinct harmonies, and dress style. Having sold tens of millions of albums, the Temptations are among the most successful groups in popular music. Featuring five male vocalists and dancers (save for brief periods with . . .

Doc 3 (Title: "Otis Williams") Otis Williams Otis Williams (born Otis Miles Jr.; October 30, 1941) is an American baritone singer. Nicknamed ""Big Daddy"", he is occasionally also a songwriter and a record producer. Williams . . . `</information>`

`<think>`I found out that David Ruffin, Otis Williams, and other singers were part of The Temptations. I need to find the main or most famous singer from the group.`</think>`

`<search>` most famous singer in The Temptations `</search>`

`<information>` Doc 1 (Title: "David Ruffin") David Ruffin David Eli Ruffin (born Davis Eli Ruffin, January 18, 1941 – June 1, 1991) was an American soul singer and musician most famous for his work as one of the lead singers of The Temptations (1964–68) during the group's ""Classic Five"" period as it was later known. He was the lead voice on such . . .

Doc 2 (Title: "Otis Williams") Otis Williams Otis Williams (born Otis Miles Jr.; October 30, 1941) is an American baritone singer. Nicknamed ""Big Daddy"", he is occasionally also a songwriter and a record producer. Williams . . .

Doc 3 (Title: "The Temptations") The Temptations The Temptations are an American vocal group who released a series of successful singles and albums with Motown Records during the 1960s and 1970s. Their work with producer Norman Whitfield, beginning with the Top 10 hit single ""Cloud Nine"" in October 1968, pioneered psychedelic soul, and was significant in the evolution of R&B and soul music. The band members are known for their choreography, distinct harmonies, and dress style. Having sold tens of millions of albums, the Temptations are among the most successful groups in popular music. Featuring five male vocalists and dancers (save for brief periods with . . . `</information>`

`<think>`I found out that the main singer of The Temptations is not explicitly mentioned.`</think>`

`<answer>` David Ruffin `</answer>`

---

Table 15: Example of hacking questions: question with non-unique answers (correct answer).

---

**Solver** answers the question raised by Proposer: *Temptations singer?*

`<think>`I need to find out which singer of the Temptations. I'll search for it.`</think>`

`<search>` Temptations singer `</search>`

`<information>` Doc 1 (Title: "Terry Weeks") Terry Weeks Terry Wayne Weeks (born December 23, 1963) is an American R&B and soul singer who is currently one of the lead singers of the legendary Motown . . .

Doc 2 (Title: "Otis Williams") Otis Williams Otis Williams (born Otis Miles Jr.; October 30, 1941) is an American baritone singer. Nicknamed ""Big Daddy"", he is occasionally also a songwriter and a record producer . . .

Doc 3 (Title: "The Temptations") The Temptations The Temptations are an American vocal group who released a series of successful singles and albums with Motown Records during the 1960s and 1970s. Their work with producer Norman Whitfield, beginning with the Top 10 hit single ""Cloud Nine"" in October 1968, pioneered psychedelic soul, and was significant in the evolution of R&B and soul music. The band members are known for their choreography, distinct harmonies, and dress style. Having sold tens of millions of albums, the Temptations are among the most successful groups in popular music. Featuring five male . . . `</information>`

`<think>`I found out that The Temptations is a vocal group. Now I need to find which singer of the Temptations.`</think>`

`<search>` who is a singer of the Temptations `</search>`

`<information>` Doc 1 (Title: "The Temptations") The Temptations The Temptations are an American vocal group who released a series of successful singles and albums with Motown Records during the 1960s and 1970s. Their work with producer Norman Whitfield, beginning with the Top 10 hit single ""Cloud Nine"" in October 1968, pioneered psychedelic soul, and was significant in the evolution of R&B and soul music. The band members are known for their choreography, distinct harmonies, and dress style. Having sold tens of millions of albums, the Temptations are among the most successful groups in popular music. Featuring five male vocalists and dancers (save for brief periods with . . .

Doc 2 (Title: "Otis Williams") Otis Williams Otis Williams (born Otis Miles Jr.; October 30, 1941) is an American baritone singer. Nicknamed ""Big Daddy"", he is occasionally also a songwriter and a record producer . . .

Doc 3 (Title: "The Temptations") Ten on either the United States Pop or R&B Albums Chart. The Temptations The Temptations are an American vocal group who released a series of successful singles and albums with Motown Records during the 1960s and 1970s. Their work with producer Norman Whitfield, beginning with the Top 10 hit single ""Cloud Nine"" in October 1968, pioneered psychedelic soul, and was significant . . . `</information>`

`<think>`I found out that The Temptations is a vocal group. Now I need to find a specific singer of the Temptations.`</think>`

`<search>` singer of The Temptations `</search>`

`<information>` Doc 1 (Title: "Otis Williams") Otis Williams Otis Williams (born Otis Miles Jr.; October 30, 1941) is an American baritone singer. Nicknamed ""Big Daddy"", he is occasionally also a songwriter and a record producer. Williams . . .

Doc 2 (Title: "The Temptations") The Temptations The Temptations are an American vocal group who released a series of successful singles and albums with Motown Records during the 1960s and 1970s. Their work with producer . . .

Doc 3 (Title: "Paul Williams (The Temptations)") Paul Williams (The Temptations) Paul Williams (July 2, 1939 – August 17, 1973) was an American baritone singer and choreographer. Williams was noted for being one of the founding members and original lead singer of the Motown group The Temptations. Along with Elbridge ""Al"" Bryant, Otis Williams , and fellow Alabamians Eddie Kendricks and Melvin Franklin, Williams was a member of The Temptations . . . `</information>`

`<think>`I found out that the singers of The Temptations include Otis Williams, Eddie Kendricks, and Paul Williams. Now I need to find a specific singer of the Temptations.`</think>`

`<search>` temptations lead singer `</search>`

`<information>` Doc 1 (Title: "Otis Williams") Otis Williams Otis Williams (born Otis Miles Jr.; October 30, 1941) is an American baritone singer. Nicknamed ""Big Daddy"", he is occasionally also a songwriter and a record producer. Williams . . .

Doc 2 (Title: "Terry Weeks") Terry Weeks Terry Wayne Weeks (born December 23, 1963) is an American R&B and soul singer who is currently one of the lead singers of the legendary Motown quintet The Temptations. Weeks was born in Birmingham, Alabama. Serving in the Air Force, including involvement with Tops In Blue for eight years. Upon his retirement in the early-1990s, Weeks began singing locally around his home . . .

Doc 3 (Title: "Terry Weeks") taking the place of Temptations lead Ali-Ollie Woodson, who was battling throat cancer. He would replace Woodson permanently in 1997, joining the Temptations lineup that included Williams, Ron Tyson, Theo Peoples and Harry McGilberry in 1996. Weeks sung co-lead on the . . . `</information>`

`<think>`I found out that the lead singer of The Temptations might be Otis Williams. Now I can provide the answer.`</think>`

`<answer>` Otis Williams `</answer>`

