# OpenReview forum: "Search Self-Play: Pushing the Frontier of Agent Capability without Supervision"
_ICLR.cc/2026/Conference — ICLR 2026 Poster_

### Official Review · Reviewer_tjcn · 2025-10-21

**Soundness:** 3
**Presentation:** 2
**Contribution:** 3
**Rating:** 4
**Confidence:** 4

**Summary:**

The paper proposes Search Self-Play (SSP), a novel reinforcement learning framework aimed at improving deep search agents without human supervision. SSP involves two roles for a single LLM: a task proposer that generates challenging search queries, and a problem solver that attempts to answer these queries using multi-turn search interactions. The framework ensures the correctness of queries by employing a retrieval-augmented generation (RAG) process to verify ground-truth answers. Experimental results demonstrate that SSP improves agent performance across various benchmarks, including NQ, TriviaQA, and HotpotQA, both in from-scratch and continuous training setups.

**Strengths:**

This paper makes a significant contribution by introducing SSP as a self-supervised training method for search agents. The originality lies in the combination of search agent roles with self-play, where the proposer and solver co-evolve. This approach represents a promising direction for training LLMs autonomously and addresses key challenges such as task synthesis and scalability. The empirical results across a diverse set of benchmarks, particularly with the SSP framework applied to various LLMs, demonstrate the method's potential for future agentic training.

**Weaknesses:**

* The method samples ground-truth answers from a pre-defined set (D) to drive self-play but does not disclose (D)’s source, size, or curation, nor its overlap with evaluation answers; this raises potential data-leakage/overfitting concerns. Please report (D)’s provenance, scale, overlap with each benchmark, and blacklist/de-dup procedures.

* Using a solver **without tools** as the verifier is an unreliable proxy for question validity: failures may reflect the verifier’s limited contextual/multi-hop reasoning (or added RAG noise) rather than flaws in the proposed question, causing valid items to be wrongly rejected.

* Evaluation relies on a single judge (Qwen2.5-32B-Instruct) with pass@1 while several tested systems are Qwen-family models, inviting family bias and prompt-format gaming; adding exact match (EM) and F1 would provide more robust evidence.

* Strong results on NQ/TriviaQA/HotpotQA/2Wiki/MuSiQue/Bamboogle notwithstanding, coverage skews to classic QA; testing on newer search benchmarks (e.g., BrowseComp and SimpleQA) would better demonstrate robustness.

**Questions:**

* Clarify how the answer set (D) is built (source, size, type distribution) and report overlap stats with each benchmark, plus de-dup/blacklist procedures.
* Add EM/F1 alongside pass@1, or use heterogeneous judges from different families, and report inter-judge agreement.
* Expand evaluation to newer search/browsing benchmarks (e.g., BrowseComp; SimpleQA if applicable).
* Why is there no reward for SSP Proposer in Figure 3(a)?

---

> ### Author Response · Authors · 2025-11-23
> **Response to Reviewer tjcn (Part 1)**
>
> ### Q1: The method samples ground-truth answers from a pre-defined set (D) to drive self-play but does not disclose (D)'s source, size, or curation, nor its overlap with evaluation answers; this raises potential data-leakage/overfitting concerns. Please report (D)'s provenance, scale, overlap with each benchmark, and blacklist/de-dup procedures.
>
> A1: Thank you for pointing out this important concern. We clarify the construction and usage of the predefined answer set $D$ as follows:
> - **The entire set $D$ is sampled exclusively from public training data.** Specifically, we construct $D$ by randomly sampling ground-truth answers from (1) the Search-R1 [1] training set (i.e., the training sets of NQ and HotpotQA), and (2) the ground-truth answers used in ARPO's released training set [2].
> - We sample **50,000 ground-truth answers** to form the final answer pool $D$.
> - **We added a new Figure 4 and a new Section A.4 in supplementary material, presenting the diversity and length distribution of $D$.** Specifically, the answers have an averaged length of 14.53. $D$ spans a broad range of topics, including People (26.3%), Time & Dates (12.9%), Geography & Places (9.2%), Music (8.4%), Sports (5.5%), Film/TV/Entertainment (4.4%), History (4.1%), Law & Politics (3.8%), Biology (3.8%), among others.
> - We compute the intersection between $D$ and all evaluation answers. Only 1,080 answers overlap. Moreover, these overlaps are predominantly non-informative lexical items such as:
>     - numbers (e.g., "0", "fourteen", "20%", "third"),
>     - dates/time expressions (e.g., "November", "four-year", "18th century"),
>     - geographic entities (e.g. "Brazil", "California", "the U.S."),
>     - generic semantic categories (e.g., "space", "singer", "heart").
>
>     **These items are extremely common and unavoidable across open-domain QA datasets, and do not constitute content-level leakage.**
>
> We will include all details in the revised appendix and open-source all data used in the paper.
>
> [1] Bowen Jin, Hansi Zeng, Zhenrui Yue, Jinsung Yoon, Sercan Arik, Dong Wang, Hamed Zamani, and Jiawei Han. Search-r1: Training llms to reason and leverage search engines with reinforcement learning. arXiv preprint arXiv:2503.09516, 2025b.
>
> [2] Guanting Dong, Hangyu Mao, Kai Ma, Licheng Bao, Yifei Chen, Zhongyuan Wang, Zhongxia Chen, Jiazhen Du, Huiyang Wang, Fuzheng Zhang, Guorui Zhou, Yutao Zhu, Ji-Rong Wen, and Zhicheng Dou. Agentic Reinforced Policy Optimization. arXiv preprint arXiv:2507.19849, 2025.
>
> ---
>
> ### Q2: Using a solver without tools as the verifier is an unreliable proxy for question validity: failures may reflect the verifier's limited contextual/multi-hop reasoning (or added RAG noise) rather than flaws in the proposed question, causing valid items to be wrongly rejected.
>
> A2: Thank you for raising this important point. We agree that a solver without tools may fail on some valid questions due to its own reasoning limitations or RAG noise.
>
> In designing the SSP framework, we have prioritized **high precision in the verification process**, even at the cost of some recall. While this means a subset of valid questions might be filtered out, it effectively ensures that only high-confidence samples enter the self-play loop. This selective mechanism helps maintain a clean and stable training environment, which we find essential for facilitating robust co-evolution between the proposer and solver.
>
> In summary, we view this trade-off as intentional: minimizing noisy or invalid questions from entering the training data supports more stable and efficient learning dynamics in the long run.
>
> ---
>
> ### Q3: Evaluation relies on a single judge (Qwen2.5-32B-Instruct) with pass@1 while several tested systems are Qwen-family models, inviting family bias and prompt-format gaming; adding exact match (EM) and F1 would provide more robust evidence.
>
> A3: Thank you for raising this concern. We address it from two perspectives:
>
> (1) **Our evaluation is not limited to Qwen2.5 series.** As shown in Table 1 of the paper, we additionally evaluate SSP on non-Qwen2.5 models, including LLaMA-3.1 and Qwen3, and observe consistent improvements across all architectures. These results demonstrate that SSP is model-agnostic and does not rely on family-specific biases or prompt-format artifacts.
>
> | Method | NQ | TriviaQA | PopQA | HotpotQA | 2Wiki | MuSiQue | Bamboogle | Avg |
> | --- | --- | --- | --- | --- | --- | --- | --- | --- |
> | **LLaMA-3.1-8B** | 50.2 | 65.2 | 45.8 | 34.6 | 19.4 | 11.4 | 30.4 | 36.7 |
> | **+ SSP** | **58.0** (+7.8) | **75.8** (+10.6) | **55.4** (+9.6) | **44.2** (+9.6) | **34.4** (+15.0) | **16.2** (+4.8) | **40.0** (+9.6) | **46.3** (+9.6) |
> | **Qwen3-8B** | 53.6 | 76.0 | 50.8 | 54.2 | 48.0 | 26.6 | 58.4 | 52.5 |
> | **+ SSP** | **56.0** (+2.4) | **78.2** (+2.2) | **55.0** (+4.2) | **58.0** (+3.8) | **51.5** (+3.5) | **28.0** (+1.4) | **67.2** (+8.8) | **56.3** (+3.8) |

---

> ### Author Response · Authors · 2025-11-23
> **Response to Reviewer tjcn (Part 2)**
>
> (2) **We have conducted EM/F1 evaluations on all relevant benchmarks.** The results are summarized in the table below:
> | Method | NQ | TriviaQA | PopQA | HotpotQA | 2Wiki | MuSiQue | Bamboogle |
> | --- | --- | --- | --- | --- | --- | --- | --- |
> |  | EM/F1 | EM/F1 | EM/F1 | EM/F1 | EM/F1 | EM/F1 | EM/F1 |
> | **Qwen2.5-7B-Instruct** | 28.6/37.5 | 50.0/59.2 | 30.8/36.1 | 27.8/40.0 | 26.4/33.0 | 11.8/18.9 | 35.2/47.6 |
> | **+ SSP** | **33.4/43.6** | **54.2/65.0** | **40.4/46.7** | **29.0/41.0** | **27.4/34.5** | **12.6/20.2** | **38.4/49.8** |
> | **Search-R1-7B** | 48.6/55.5 | 62.0/71.6 | 52.2/56.7 | 42.8/56.2 | 38.8/46.6 | 21.0/30.1 | 42.4/54.9 |
> | **+ SSP** | **48.4/56.1** | **64.0/73.5** | **53.0/57.7** | **45.6/58.5** | **40.0/47.6** | **21.0/30.9** | **48.0/59.2** |
> | **ZeroSearch-7B** | 36.4/47.4 | 54.2/64.0 | 42.4/49.8 | 29.4/41.0 | 29.4/38.5 | 11.6/19.5 | 35.2/41.6 |
> | **+ SSP** | **38.8/49.9** | **55.8/66.5** | **45.0/52.4** | **29.8/41.8** | **30.6/40.9** | **13.4/21.7** | **36.8/45.0** |
> | **R-Search-7B** | 38.4/47.6 | 59.4/67.7 | 46.6/52.7 | 39.8/52.3 | 50.2/56.0 | 21.0/29.6 | 40.0/53.4 |
> | **+ SSP** | **41.8/50.2** | **61.6/70.3** | **49.2/55.1** | **38.2/51.6** | **50.2/55.9** | **22.0/32.0** | **43.2/55.9** |
>
> These additional metrics corroborate our main findings: SSP yields significant performance gains under different evaluation protocols, further validating the effectiveness and generality of our approach.
>
> ---
> ### Q4: Strong results on NQ/TriviaQA/HotpotQA/2Wiki/MuSiQue/Bamboogle notwithstanding, coverage skews to classic QA; testing on newer search benchmarks (e.g., BrowseComp and SimpleQA) would better demonstrate robustness.
>
> A4: We appreciate the reviewer's suggestion to include newer search benchmarks such as BrowseComp and SimpleQA. We address this point as follows:
>
> (1) Benchmark selection and comparability.
>
> Existing work in this line of research including Search-R1 [1], ZeroSearch [2], and R-Search [3] also does not report results on BrowseComp or SimpleQA. One important reason is that benchmarking on them usually requires online web access, making its evaluation dependent on a live web retriever. This introduces substantial variance across time, regions, and retriever configurations, **making it difficult to ensure fair, reproducible, and comparable evaluation against prior work**.
>
> (2) Robustness across diverse QA settings.
>
> Although we focus on the classic open-domain QA suite, these datasets cover a wide spectrum of multi-hop reasoning, factual lookup, and search-intensive tasks. SSP shows strong and consistent gains across all of them, providing solid evidence of robustness.
>
> (3) BrowseComp result.
>
> **We performed additional evaluation on the BrowseComp benchmark using an in-house online web retriever.** However, as this retrieval system is not publicly available, the results may be difficult for the research community to reproduce. Additionally, the outcomes may not be directly comparable with those reported in other papers. Below are our experimental results:
> | Method | BrowseComp |
> | --- | --- |
> | **Qwen2.5-32B-Instruct** | 0.4 |
> | **+ SSP** | 9.7 |
>
> The results show that SSP brings a substantial performance gain on this benchmark. It further validates that SSP enhances LLM agent capabilities on complex tasks under online web retrieval conditions.
>
> [1] Bowen Jin, Hansi Zeng, Zhenrui Yue, Jinsung Yoon, Sercan Arik, Dong Wang, Hamed Zamani, and Jiawei Han. Search-r1: Training llms to reason and leverage search engines with reinforcement learning. arXiv preprint arXiv:2503.09516, 2025b.
>
> [2] Hao Sun, Zile Qiao, Jiayan Guo, Xuanbo Fan, Yingyan Hou, Yong Jiang, Pengjun Xie, Fei Huang, and Yan Zhang. Zerosearch: Incentivize the search capability of llms without searching. arXiv preprint arXiv:2505.04588, 2025a.
>
> [3] Qingfei Zhao, Ruobing Wang, Dingling Xu, Daren Zha, and Limin Liu. R-search: Empowering llm reasoning with search via multi-reward reinforcement learning. arXiv preprint arXiv:2506.04185, 2025.
>
> ---
> ### Q5: Why is there no reward for SSP Proposer in Figure 3(a)?
>
> A5: In our framework, the SSP Proposer and Solver are modeled as engaged in a min-max game. Since the Proposer's objective is to maximize the Solver's difficulty, its reward is derived from the Solver's performance by Equation (5) in the paper. Plotting both would be redundant. Therefore, to avoid redundancy and maintain clarity in Figure 3(a), we chose to explicitly plot only the Solver's reward.

---

> ### Comment · Area_Chair_GuGk · 2025-11-25
> **Please participate in discussions with authors and other reviewers asap**
>
> Please ensure you are actively participating in the discussion with authors.
>
> Additionally, I strongly encourage you to read the other reviews and discuss with your fellow reviewers. It is vital to compare perspectives and raise any remaining concerns now to give the authors a fair opportunity to respond.
>
> Based on these interactions, please update your reviews and finalize your decisions.
>
> Best, AC

---

### Official Review · Reviewer_TBC9 · 2025-10-31

**Soundness:** 3
**Presentation:** 3
**Contribution:** 3
**Rating:** 6
**Confidence:** 4

**Summary:**

The paper introduces Search Self-Play (SSP), a reinforcement learning framework where LLM takes on two alternating roles: a “proposer” that creates challenging, search-intensive QA tasks, and a “solver” that attempts to answer them. To ensure the generated questions are valid and have a unique, verifiable answer, a RAG-based verification gate is used, which relies on the proposer’s retrieved documents to validate the questions before adversarial training begins. The solver is trained using GRPO, while the proposer is optimized with REINFORCE. Experiments across seven open-domain QA benchmarks and various model architectures and sizes consistently demonstrate that SSP delivers significant performance improvements.

**Strengths:**

1. This work is novel in its application of self-play to agentic search.

2. The paper provides precise definitions of constraints and rewards, fully discloses prompts and hyperparameters, and includes training curves that illustrate the co-evolution of proposer and solver.

3. By reducing reliance on annotated agentic data, SSP achieves consistent, substantial improvements across different model architectures and scales, paving a practical path toward scalable, unsupervised training for agentic tasks.

**Weaknesses:**

1. Please discuss training stability under the adversarial/cooperative setup. Does the min–max dynamic cause frequent collapses or mode drift? How easy is convergence in practice? Please provide stability evidence (multiple seeds, valid-question rate over time, reward variance) and, if available, theoretical or empirical guarantees that prevent reward hacking and proposer entropy explosions.

2. Please report end-to-end compute and cost: total wall-clock, GPU hours, tokens processed, search calls, and energy. If SSP is substantially more resource-intensive, justify why not to to paid human annotation via crowd platforms.

3. Clarify the rationale for using REINFORCE as the proposer’s advantage estimator and GRPO for the solver. Why is this asymmetric choice preferable, and how do performance/variance trade-offs compare to alternatives (e.g., PPO, GPRO, REINFORCE)?

4. Evaluation heavily on a single LLM-as-judge and 500-sample subsets, which may introduce bias and variance.

**Questions:**

5. Provide results using real-world web content (e.g., Bing/Google APIs) and different corpora/embeddings retrievers.

6. Improve Figure 2 to more clearly reflect the pipeline. The current Figure obscures the implementation details. Consider redrawing it to mirror Algorithm 1 step-by-step (proposer rollout → rule checks → RAG verification with noise → solver rollouts → rewards/updates).

---

> ### Author Response · Authors · 2025-11-23
> **Response to Reviewer TBC9 (Part 1)**
>
> ### Q1: Please discuss training stability under the adversarial/cooperative setup. Does the min–max dynamic cause frequent collapses or mode drift? How easy is convergence in practice? Please provide stability evidence (multiple seeds, valid-question rate over time, reward variance) and, if available, theoretical or empirical guarantees that prevent reward hacking and proposer entropy explosions.
>
> A1: We acknowledge that adversarial training initially presents significant challenges, as the proposer can easily exploit reward hacking strategies. However, our framework incorporates multiple mechanisms specifically designed to prevent these common failure modes:
>
> 1. **RAG Verification Constraint**: The cooperative verification requirement prevents the proposer from generating unsolvable or incorrect questions, eliminating a major source of reward hacking. Every question must be answerable given the proposer's search results, ensuring the adversarial dynamic remains productive.
> 2. **Replay Buffer Regularization**: The periodic reset strategy (every 10 steps) prevents both proposer stagnation and solver overfitting, maintaining healthy co-evolution dynamics.
>
> **Empirical Stability Evidence:**
>
> - **Figure 3** provides direct evidence of training stability across different configurations. Our complete SSP framework shows remarkably stable training dynamics—the solver's reward curve demonstrates relatively smooth progression with low variance, contrasting sharply with the erratic behavior observed in fixed-opponent baselines. **Figure 5** and **Figure 6** in the Appendix provide additional insights into the training dynamics of both proposer and solver, further revealing the relative stability of the training process. We also observe that due to our constraints of 8k maximum context length and up to 10 tool calls, the solver's improvement on the benchmarks gradually slows down in later stages of training.
>
> - **Figure 6(b)** provides crucial evidence of training stability through the valid question rate over time. The proposer initially struggles with proper question formatting and verification requirements (near 0% success rate), but demonstrates steady improvement throughout training, reaching approximately 50% by the end. This consistent upward trend indicates stable learning dynamics without the oscillations or collapses typical of unstable adversarial training.
>
> The combination of RAG verification constraints, balanced replay buffer strategies, and progressive improvement in question quality demonstrates that our SSP framework achieves stable co-evolutionary training without the common pitfalls of adversarial setups.
>
> ---
> ### Q2: Please report end-to-end compute and cost: total wall-clock, GPU hours, tokens processed, search calls, and energy. If SSP is substantially more resource-intensive, justify why not to to paid human annotation via crowd platforms.
>
> A2:  Below, we provide a detailed breakdown of the resources consumed in one of our main experiments and justify why SSP remains a cost-effective and scalable alternative to paid human annotation.
>
> **1. Computational Cost Breakdown:**
>
> - **Total Wall-clock Time:** Approximately 21 hours.
> - **Total GPU Hours:** Approximately 285 hours.
> - **Total Tokens Processed:** Approximately $1.28\times 10^8$ tokens.
> - **Total Search Calls:** Approximately $2.5 \times 10^5$ calls.
> - **Estimated Energy Consumption:** Approximately 115 kWh.
>
> **2. SSP vs. Human Annotation:**
>
> We argue that SSP is more scalable in the long run for the following reasons:
>
> - **Towards Automated Search Agent:** The core objective of our work is to explore a **fully automated** self-play learning paradigm. Relying on human annotation would contradict this fundamental goal. The costs reported here are for building this autonomous agent capability, which we believe is a valuable direction for the community.
> - **Quality and Complexity Control:** Human annotation for complex, knowledge-intensive questions (as in our setting) is not only expensive but also suffers from quality inconsistency and scalability bottlenecks. SSP allows control over question difficulty, which is difficult to achieve reliably with crowd-sourcing.
> - **Scalability and Reusability:** Once trained, the proposer and solver models can generate and answer new questions at less marginal cost. In contrast, human annotation requires continuous payment for each new batch of questions or each distribution shift, which becomes prohibitively expensive for large-scale or iterative research.
>
> In summary, we view the computational cost of SSP as an investment in a scalable, automated framework that avoids the recurring expenses and limitations of human annotation, especially for complex and evolving search tasks.

---

> ### Author Response · Authors · 2025-11-23
> **Response to Reviewer TBC9 (Part 2)**
>
> ### Q3: Clarify the rationale for using REINFORCE as the proposer’s advantage estimator and GRPO for the solver. Why is this asymmetric choice preferable, and how do performance/variance trade-offs compare to alternatives (e.g., PPO, GPRO, REINFORCE)?
>
> A3: We conducted additional experiments to investigate different RL algorithm combinations, and this design decision is based on both performance gains and computational efficiency considerations.
>
> As shown in the table below, using GRPO for both proposer and solver (GRPO--GRPO configuration) does achieve the highest average accuracy (50.9), slightly outperforming our default RF--GRPO configuration (49.5). **However, this improvement comes at a substantial computational cost: the per-step generation time increases from 83.4 seconds to 504.4 seconds—approximately 6× slower.** The performance gains are modest across most datasets (+1.2 on NQ, +2.0 on PopQA, +3.0 on 2Wiki, +1.2 on MuSiQue, +2.4 on Bamboogle), with even a small drop on TriviaQA (-0.6).
>
> | Proposer | Solver | NQ | TriviaQA | PopQA | HotpotQA | 2Wiki | MuSiQue | Bamboogle | Avg. | T/step(s) |
> | --- | --- | --- | --- | --- | --- | --- | --- | --- | --- | --- |
> | RF | GRPO | 54.8 | 73.4 | 51.8 | 51.8 | 38.8 | 21.2 | 54.4 | 49.5 | 83.4 |
> | RF | RF | 53.0 | 71.2 | 53.6 | 46.2 | 28.0 | 13.2 | 28.8 | 42.0 | 9.1 |
> | GRPO | RF | 51.2 | 69.2 | 49.8 | 47.6 | 37.0 | 21.6 | 48.8 | 46.5 | 50.1 |
> | GRPO | GRPO | 56.0 | 72.8 | 53.8 | 52.4 | 41.8 | 22.4 | 56.8 | 50.9 | 504.4 |
>
> *Table: Evaluation results and training time comparisons on different RL algorithms for proposers and solvers, where RF denotes REINFORCE.*
>
> Interestingly, when we examine other configurations, they all underperform compared to our default RF--GRPO setting. For instance, using GRPO for the proposer with RF for the solver leads to performance degradation (46.5), while the RF--RF pairing also shows inferior results (42.0). This demonstrates that our default configuration provides the optimal balance for the SSP framework.
>
> **Given that the GRPO--GRPO configuration requires 6× more computational resources for only marginal gains (1.4 points), the RF--GRPO combination provides the optimal trade-off between effectiveness and efficiency for practical training budgets**. This is why we chose it as our default configuration.
>
> ---
> ### Q4: Evaluation heavily on a single LLM-as-judge and 500-sample subsets, which may introduce bias and variance.
>
> A4: To address these concerns, we provide clarification from two key perspectives:
>
> **(1) Sampling Strategy and Comparability**
>
> Following established practices in this research area, existing works including ZeroSearch [1], R-Search [2], ASearcher [3], and Atom-Searcher [4] all employ sampling-based evaluation rather than full dataset evaluation. We adopt the same 500-sample subset approach to ensure fair and comparable evaluation against prior work while managing computational costs. Importantly, **all methods are evaluated on identical sampled test sets**, ensuring relative fairness in our comparisons. We will open-source our complete evaluation datasets upon paper acceptance to enable full reproducibility.
>
> **(2) Multiple Evaluation Metrics Beyond LLM-as-Judge**
>
> To address potential bias from relying solely on LLM-as-judge, we have conducted comprehensive evaluations using both Exact Match (EM) and F1 scores across all benchmarks. These additional metrics corroborate our main findings:
>
> | Method | NQ (EM/F1) | TriviaQA (EM/F1) | PopQA (EM/F1) | HotpotQA (EM/F1) | 2Wiki (EM/F1) | MuSiQue (EM/F1) | Bamboogle (EM/F1) |
> | --- | --- | --- | --- | --- | --- | --- | --- |
> | Qwen2.5-7B-Instruct | 28.6/37.5 | 50.0/59.2 | 30.8/36.1 | 27.8/40.0 | 26.4/33.0 | 11.8/18.9 | 35.2/47.6 |
> | **+ SSP** | **33.4/43.6** | **54.2/65.0** | **40.4/46.7** | **29.0/41.0** | **27.4/34.5** | **12.6/20.2** | **38.4/49.8** |
>
> These results demonstrate that SSP yields consistent performance gains across different evaluation protocols (LLM-as-judge, EM, and F1), validating the robustness of our findings beyond any single evaluation metric. The convergent evidence from multiple metrics and model families further strengthens confidence in our approach's effectiveness.
>
> [1] Hao Sun, Zile Qiao, Jiayan Guo, Xuanbo Fan, et al. Zerosearch: Incentivize the search capability of llms without searching. arXiv preprint arXiv:2505.04588, 2025a.
>
> [2] Qingfei Zhao, Ruobing Wang, Dingling Xu, et al. R-search: Empowering llm reasoning with search via multi-reward reinforcement learning. arXiv preprint arXiv:2506.04185, 2025.
>
> [3] Jiaxuan Gao, Wei Fu, Minyang Xie, et al. Beyond ten turns: Unlocking long-horizon agentic search with large-scale asynchronous rl. arXiv preprint arXiv:2508.07976, 2025a.
>
> [4] Yong Deng, Guoqing Wang, Zhenzhe Ying, et al. Atom-Searcher: Enhancing Agentic Deep Research via Fine-Grained Atomic Thought Reward, August 2025b. arXiv:2508.12800.

---

> ### Author Response · Authors · 2025-11-23
> **Response to Reviewer TBC9 (Part 3)**
>
> ### Q5: Provide results using real-world web content (e.g., Bing/Google APIs) and different corpora/embeddings retrievers.
>
> A5: We appreciate the reviewer's suggestion to evaluate SSP with real-world web content and diverse retrieval systems. We address this point as follows:
>
> **(1) Reproducibility and Fair Comparison Considerations**
>
> Similar to our response regarding newer benchmarks, existing works in this research area including Search-R1 [1], ZeroSearch [2], and R-Search [3] primarily use static, controlled retrieval environments (e.g., Wikipedia corpus with local retrievers) for evaluation. Using live web APIs (Bing/Google) introduces significant evaluation challenges:
>
> - **Temporal variance**: Web content changes over time, making results non-reproducible
> - **Geographic variance**: API responses vary by region and access patterns
> - **Rate limiting and cost**: Live APIs impose usage constraints that affect experimental scalability
> - **Fair comparison**: Results become incomparable with prior work due to different retrieval conditions
>
> **(2) Controlled Evaluation for Method Validation**
>
> Our choice of a static Wikipedia 2018 corpus with local E5 retrieval serves multiple purposes:
>
> - **Reproducibility**: Ensures consistent evaluation conditions across experiments
> - **Fair baseline comparison**: All methods operate under identical retrieval constraints
> - **Method isolation**: Allows us to isolate the effectiveness of SSP from retrieval system variations
> - **Computational feasibility**: Enables large-scale training without API cost constraints
>
> **(3) Retrieval System Robustness Evidence**
>
> While we use a single retrieval setup for controlled comparison, our results demonstrate robustness across diverse question types and reasoning patterns within this framework. The consistent improvements across seven different benchmarks (covering factual QA, multi-hop reasoning, and complex search tasks) provide strong evidence that SSP's benefits are not tied to specific retrieval characteristics.
>
> **(4) Future Extensions**
>
> We acknowledge that real-world deployment would benefit from evaluation with live web APIs and diverse retrievers. **We performed additional evaluation on the BrowseComp benchmark using an in-house online web retriever.** However, as this retrieval system is not publicly available, the results may be difficult for the research community to reproduce. Additionally, the outcomes may not be directly comparable with those reported in other papers. Below are our experimental results:
>
> | Method | BrowseComp |
> | --- | --- |
> | **Qwen2.5-32B-Instruct** | 0.4 |
> | **+ SSP** | 9.7 |
>
> The results show that SSP brings a substantial performance gain. It further validates that SSP enhances LLM agent capabilities on complex tasks under online web retrieval conditions.
>
> If the paper is accepted, we are committed to conducting supplementary experiments with different retrieval systems and will include these results with appropriate caveats about comparability in future work or extended versions.
>
> [1] Bowen Jin, Hansi Zeng, Zhenrui Yue, Jinsung Yoon, Sercan Arik, Dong Wang, Hamed Zamani, and Jiawei Han. Search-r1: Training llms to reason and leverage search engines with reinforcement learning. arXiv preprint arXiv:2503.09516, 2025b.
>
> [2] Hao Sun, Zile Qiao, Jiayan Guo, Xuanbo Fan, Yingyan Hou, Yong Jiang, Pengjun Xie, Fei Huang, and Yan Zhang. Zerosearch: Incentivize the search capability of llms without searching. arXiv preprint arXiv:2505.04588, 2025a.
>
> [3] Qingfei Zhao, Ruobing Wang, Dingling Xu, Daren Zha, and Limin Liu. R-search: Empowering llm reasoning with search via multi-reward reinforcement learning. arXiv preprint arXiv:2506.04185, 2025.
>
> ---
> ### Q6: Improve Figure 2 to more clearly reflect the pipeline. The current Figure obscures the implementation details. Consider redrawing it to mirror Algorithm 1 step-by-step (proposer rollout → rule checks → RAG verification with noise → solver rollouts → rewards/updates).
>
> A6:  We agree that a step-by-step visualization mirroring Algorithm 1 would better elucidate the implementation details. According to your suggestions, we will redraw Figure 2 to provide a clearer, step-by-step illustration of the pipeline as outlined in Algorithm 1, ensuring the implementation details are more explicitly shown in the final version.

---

> ### Comment · Area_Chair_GuGk · 2025-11-25
> **Please participate in discussions with authors and other reviewers asap**
>
> Please ensure you are actively participating in the discussion with authors.
>
> Additionally, I strongly encourage you to read the other reviews and discuss with your fellow reviewers. It is vital to compare perspectives and raise any remaining concerns now to give the authors a fair opportunity to respond.
>
> Based on these interactions, please update your reviews and finalize your decisions.
>
> Best, AC

---

### Official Review · Reviewer_BnWv · 2025-11-01

**Soundness:** 3
**Presentation:** 3
**Contribution:** 3
**Rating:** 6
**Confidence:** 4

**Summary:**

This work introduces Search Self-Play (SSP), a reinforcement learning framework for training deep search agents without human supervision. The approach uses a single LLM in dual roles: as a question proposer that generates search queries with verifiable ground-truth answers, and as a problem solver that attempts to answer these questions through multi-turn search interactions. The key component is a RAG-based verification mechanism that ensures the proposed questions are answerable given the proposer's retrieved documents, preventing reward hacking. The authors formulate this problem as a cooperative competition scenario, where the proposer and solver collaborate during the RAG-based verification but compete through the proposer generating increasingly difficult questions and the solver learning to get better at answering difficult questions.

Experiments demonstrate consistent improvements across seven QA benchmarks. The work addresses a significant bottleneck in RLVR (Reinforcement Learning with Verifiable Rewards): the need for large-scale, human-curated question-answer pairs, through a self-sustaining closed learning loop.

However, the evaluation is limited to a Wikipedia-based retrieval with a 2018 corpus, raising questions about generalization to modern web search scenarios. Note that contemporary works have also used a similar evaluation strategy in the past. The paper lacks analysis on question quality evolution, computational costs, and comparison with recent synthetic data generation methods. The RAG verification mechanism increases the computational complexity of the method, warranting a comparison of training costs with prior works.

**Strengths:**

1. Novel Problem Formulation: The paper addresses a fundamental challenge in agentic RL training, Data Scarcity, through an elegant self-play mechanism that grounds question generation in external search rather than relying solely on the model's internal knowledge.
2. Thorough Ablation Studies: The paper provides valuable insights through ablations on training schemes (co-evolution vs fixed-opponent), batch completion strategies, RAG verification configurations, and reward design, with detailed training dynamics analysis.
3. Strong Empirical Results: The proposed method shows consistent improvements across all seven benchmarks, with particularly impressive gains on base models.
4. Robust Verification Mechanism: The RAG-based verification with noisy document injection is a principled approach to prevent reward hacking and ensure question quality, though it adds computational overhead.

**Weaknesses:**

1. Limited Evaluation Setting: All experiments use a local E5 retriever with a Wikipedia 2018 corpus, which is significantly more constrained than the actual web search scenarios. The paper doesn't evaluate whether SSP-trained agents generalize to real web search or more recent knowledge bases. The recently released BrowseComp dataset is a good candidate for evaluation.
2. Lack of question quality analysis: While the paper demonstrates that the proposer generated increasingly difficult questions (as seen by declicing solver rewards in Figure 3a), there is no systematic analysis of question quality, diversity, or naturalness. It is unclear if SSP can generate diverse questions or if it leads to a narrow distribution of question types.
3. Missing Computational Cost analysis: The paper doesn't report training time, computational costs, or the overhead introduced by RAG verification and dynamic sampling. Given that invalid questions require re-generation and each question needs verification, the actual compute requirements could be substantially higher than standard RLVR, but this is not quantified.
4. Insufficient Comparison with Recent Synthetic Data Methods: While the paper mentions WebSailor (Li et al., 2025) and WebDancer (Wu et al., 2025) in related work, there's no empirical comparison. These methods also generated synthetic training data offline; a direct comparison would clarify SSP's advantages.
5. Ground Truth Answer Set: The paper states that ground truth answers are drawn from a pre-defined set D, but never specifies what this set contains, its size, or how it was constructed. This is a critical detail as it determines the scope and diversity of possible training questions.
6. Limited Analysis of Long-Term training dynamics: Training runs for only 150-200 steps. The paper doesn't investigate whether performance saturates and whether question difficulty continues to increase appropriately.

**Questions:**

1. Since the method is formulated as a co-evolution game, is there an equilibrium that can be reached here? What type of equilibrium exists in this setting?
2. What is the composition of the pre-defined answer set D? How was it constructed, and does it cover diverse answers or long-form answers?
3. What is the computational overhead of SSP compared to other works? Specifically, what percentage of generated questions pass the RAG verification as the training progresses, and how much additional compute is required for dynamic resampling/replay buffer?
4. Can you provide a quantitative analysis of question diversity over training, e.g., topic distribution and question type distribution? I am particularly interested in the question of diversity when the solver accuracy decreases. Is it a case of reward hacking from the perspective of the proposer?
5. How does SSP compare empirically to recent offline synthetic data generation methods (WebSailor and WebDancer) in terms of both final performance and data efficiency?
6. Could you evaluate the method on newer, more challenging datasets such as BrowseComp?
7. Do you observe any correlation between the proposer's search trajectory length and the solver's success rate? In other words, do more complex proposer searches lead to harder questions?
8. Curiosity Question: Do you have any insights on why dynamic resampling performs worse than Replay Buffer? Intuitively, Dynamic Resampling should be performing better even if it incurs higher computational costs. Please correct me if I am wrong here.

---

> ### Author Response · Authors · 2025-11-23
> **Response to Reviewer BnWv(Part 1)**
>
> ### Q1: Since the method is formulated as a co-evolution game, is there an equilibrium that can be reached here? What type of equilibrium exists in this setting?
>
> A1: Thank you for this insightful theoretical question about equilibrium in our co-evolution framework.
>
> - From a game-theoretic perspective, our SSP framework does admit a Nash equilibrium where neither the proposer nor the solver can unilaterally improve their performance by changing strategies. The proposer reaches a state where it cannot further increase solver failure rates, while the solver cannot further improve its success rates given the proposer's question distribution. This equilibrium is guaranteed to exist because our framework operates within a finite strategy space—the finite vocabulary, bounded context lengths, and discrete action spaces ensure that mixed-strategy Nash equilibria exist according to fundamental game theory theorems.
> - In our experimental setup, however, the equilibrium behavior is shaped by several practical constraints. We set a maximum context length of 8K tokens and limit search operations to 10 rounds, which naturally imposes an upper bound on the difficulty of questions that the proposer can generate. Once the proposer reaches this complexity ceiling, it cannot create arbitrarily harder problems, leading the solver to gradually converge toward a stable performance level on this bounded question distribution. Eventually, this can result in the solver overfitting to the proposer's constrained question space, as evidenced by the training dynamics shown in Figure 4 where solver rewards initially improve but then stabilize. This bounded equilibrium ensures system stability and prevents the potential instability of unbounded adversarial games, though it may limit the extent of continued improvement once computational and complexity limits are reached.
>
> ### Q2：What is the composition of the pre-defined answer set D? How was it constructed, and does it cover diverse answers or long-form answers?
>
> A2: We clarify the construction and usage of the predefined answer set $D$ as follows:
>
> - **The entire set $D$ is sampled exclusively from public training data.** Specifically, we construct $D$ by randomly sampling ground-truth answers from (1) the Search-R1 [1] training corpus (i.e., the training sets of NQ and HotpotQA), and (2) the ground-truth answers used in ARPO's released training set [2].
> - **We sample 50,000 ground-truth answers to form the final answer pool $D$.**
> - **We added a new Figure 4 and a new Section A.4 in supplementary material, presenting the diversity and length distribution of $D$.** The answer set spans a broad range of topics, including People (26.3%), Time & Dates (12.9%), Geography & Places (9.2%), Music (8.4%), Sports (5.5%), Film/TV/Entertainment (4.4%), History (4.1%), Law & Politics (3.8%), Biology (3.8%), among others
> - In this search scenario, the answers primarily consist of relatively short answers with an averaged length of 14.53. We have not specifically considered extremely long answers as they are more difficult to verify the correctness in RLVR.
>
> [1] Bowen Jin, Hansi Zeng, Zhenrui Yue, Jinsung Yoon, Sercan Arik, Dong Wang, Hamed Zamani, and Jiawei Han. Search-r1: Training llms to reason and leverage search engines with reinforcement learning. arXiv preprint arXiv:2503.09516, 2025b.
>
> [2] Guanting Dong, Hangyu Mao, Kai Ma, Licheng Bao, Yifei Chen, Zhongyuan Wang, Zhongxia Chen, Jiazhen Du, Huiyang Wang, Fuzheng Zhang, Guorui Zhou, Yutao Zhu, Ji-Rong Wen, and Zhicheng Dou. Agentic Reinforced Policy Optimization. arXiv preprint arXiv:2507.19849, 2025.

---

> > ### Author Response · Authors · 2025-11-23
> > **Response to Reviewer BnWv(Part 2)**
> >
> > ### Q3：What is the computational overhead of SSP compared to other works? Specifically, what percentage of generated questions pass the RAG verification as the training progresses, and how much additional compute is required for dynamic resampling/replay buffer?
> >
> > A3: We clarify these questions as follows:
> >
> > - **Computational Overhead Analysis:** Computationally, SSP requires both proposer and solver sampling and updates, making it more expensive than training only the solver. However, SSP eliminates the need for external data construction, resulting in lower overall costs compared to supervised approaches that require extensive human annotation or complex synthetic data pipelines.
> > - **RAG Verification Pass Rate:** Figure 6(b) shows the question validation success rate throughout training. The pass rate steadily improves from near 0% to approximately 50% by the end of training, indicating that the proposer learns to generate higher-quality questions that successfully pass verification constraints.
> > - **Dynamic Resampling/Replay Buffer Efficiency:** Contrary to intuition, these strategies actually reduce computational costs rather than increase them. For dynamic resampling, we do not repeatedly sample the same prompt - instead, we directly sample the next batch of prompts until all questions are valid. This avoids wasted solver computation on invalid questions. The replay buffer strategy is even more efficient, directly filling slots with previously validated questions from earlier steps, achieving better performance with lower computational overhead than dynamic resampling.
> > ---
> > ### Q4 Can you provide a quantitative analysis of question diversity over training, e.g., topic distribution and question type distribution? I am particularly interested in the question of diversity when the solver accuracy decreases. Is it a case of reward hacking from the perspective of the proposer?
> >
> > A4: We provide comprehensive quantitative analysis addressing your concerns.
> >
> > **Quantitative Diversity Analysis (Figure 6 in the Appendix):** We conduct multi-dimensional diversity tracking:
> >
> > - **Topic Distribution (Figure 6d):**  Using LDA clustering, we identify 8 major topic clusters with each maintaining 10-15% representation throughout training, showing stable diversity.
> > - **Question Complexity (Figure 6c):**  Difficulty scores increase progressively , indicating genuine complexity growth rather than exploitation.
> >
> > **Solver Accuracy Decreases ≠ Reward Hacking:**  The key insight is that solver accuracy drops represent **effective curriculum learning**, not hacking. Figure 3a shows solver reward initially rises then controllably declines as the proposer learns to generate harder questions, which is the intended self-play dynamic. Crucially, Figure 6d shows topic diversity remains stable even when solver accuracy decreases, whereas reward hacking would cause topic collapse toward exploitable domains.
> >
> > The solver accuracy decreases represent successful adaptive curriculum learning where the proposer provides increasingly sophisticated challenges, driving genuine co-evolutionary improvement rather than reward hacking.
> >
> > ### Q5 How does SSP compare empirically to recent offline synthetic data generation methods (WebSailor and WebDancer) in terms of both final performance and data efficiency?
> >
> > A5: WebSailor and WebDancer both rely on online search engines for evaluation, whereas we validate SSP's effectiveness using a local retriever setup that ensures reproducibility and stability. This difference makes direct performance comparisons challenging, but our local setup provides more controlled and reliable evaluation conditions. WebSailor and WebDancer did not report their data efficiency, which may be significantly influenced by synthetic data scale, initial knowledge-graph scale, hardware, search engines, and their LLM-based multi-step inject-then-fuzz pipeline. This also makes direct efficiency comparisons challenging.

---

> > > ### Author Response · Authors · 2025-11-23
> > > **Response to Reviewer BnWv(Part 3)**
> > >
> > > ### Q6 Could you evaluate the method on newer, more challenging datasets such as BrowseComp?
> > >
> > > A6: We appreciate the reviewer's suggestion to include newer search benchmarks such as BrowseComp. We address this point as follows:
> > >
> > > (1) Benchmark selection and comparability.
> > >
> > > Existing work in this line of research including Search-R1 [1], ZeroSearch [2], and R-Search [3] also does not report results on BrowseComp. One important reason is that benchmarking on BrowseComp usually requires online web access, making its evaluation dependent on a live web retriever. This introduces substantial variance across time, regions, and retriever configurations, **making it difficult to ensure fair, reproducible, and comparable evaluation against prior work**.
> > >
> > > (2) Robustness across diverse QA settings.
> > >
> > > Although we focus on the classic open-domain QA suite, these datasets cover a wide spectrum of multi-hop reasoning, factual lookup, and search-intensive tasks. SSP shows strong and consistent gains across all of them, providing solid evidence of robustness.
> > >
> > > (3) BrowseComp result.
> > >
> > > **We performed additional evaluation on the BrowseComp benchmark using an in-house online web retriever.** However, as this retrieval system is not publicly available, the results may be difficult for the research community to reproduce. Additionally, the outcomes may not be directly comparable with those reported in other papers. Below are our experimental results:
> > >
> > > | Method | BrowseComp |
> > > | --- | --- |
> > > | **Qwen2.5-32B-Instruct** | 0.4 |
> > > | **+ SSP** | 9.7 |
> > >
> > > [1] Bowen Jin, Hansi Zeng, Zhenrui Yue, Jinsung Yoon, Sercan Arik, Dong Wang, Hamed Zamani, and Jiawei Han. Search-r1: Training llms to reason and leverage search engines with reinforcement learning. arXiv preprint arXiv:2503.09516, 2025b.
> > >
> > > [2] Hao Sun, Zile Qiao, Jiayan Guo, Xuanbo Fan, Yingyan Hou, Yong Jiang, Pengjun Xie, Fei Huang, and Yan Zhang. Zerosearch: Incentivize the search capability of llms without searching. arXiv preprint arXiv:2505.04588, 2025a.
> > >
> > > [3] Qingfei Zhao, Ruobing Wang, Dingling Xu, Daren Zha, and Limin Liu. R-search: Empowering llm reasoning with search via multi-reward reinforcement learning. arXiv preprint arXiv:2506.04185, 2025.
> > >
> > > ---
> > >
> > > ### Q7 Do you observe any correlation between the proposer's search trajectory length and the solver's success rate? In other words, do more complex proposer searches lead to harder questions?
> > >
> > > A7: Our additional experiments presented in Appendix B.3 reveal a clear positive correlation between proposer search complexity and question difficulty. As shown in Figure 6a and 6c in Appendix B.3, both the proposer's search count and question difficulty increase progressively throughout training. This increasing question difficulty directly explains the solver's reward dynamics observed in Figure 3a, where the solver's success rate decreases over time as the proposer learns to generate harder questions through more complex search trajectories. The solver reward shows an initial increase followed by a controlled decline, which represents the expected co-evolutionary pattern of adaptive curriculum learning rather than training failure. This correlation demonstrates that the proposer's enhanced search sophistication directly translates into more challenging questions, creating the intended adversarial pressure that drives continuous improvement in the SSP framework.
> > >
> > > ---
> > >
> > > ### Q8:Curiosity Question: Do you have any insights on why dynamic resampling performs worse than Replay Buffer? Intuitively, Dynamic Resampling should be performing better even if it incurs higher computational costs. Please correct me if I am wrong here.
> > >
> > > A8: Your intuition is understandable, but there's a key training efficiency factor that explains this counterintuitive result. The fundamental difference lies in **data utilization efficiency**. In our dynamic resampling approach, the proposer continuously generates fresh questions from new batch prompts until we obtain a complete batch of valid questions. While this ensures novelty, it means that each valid question is only seen by the solver for **exactly one training epoch** before being discarded.
> > >
> > > In contrast, the replay buffer strategy allows certain high-quality questions to be **revisited multiple times** across different training iterations. This is essentially equivalent to giving the solver **multiple epochs** to learn from the proposer's generated problems. This repeated exposure enables the solver to extract richer learning signals from each carefully crafted question, leading to more thorough understanding and better generalization.

---

> ### Comment · Area_Chair_GuGk · 2025-11-25
> **Please participate in discussions with authors and other reviewers asap**
>
> Please ensure you are actively participating in the discussion with authors.
>
> Additionally, I strongly encourage you to read the other reviews and discuss with your fellow reviewers. It is vital to compare perspectives and raise any remaining concerns now to give the authors a fair opportunity to respond.
>
> Based on these interactions, please update your reviews and finalize your decisions.
>
> Best, AC

---

> ### Comment · Reviewer_BnWv · 2025-11-26
> **Response to Authors**
>
> Thank you for providing detailed responses to the questions. Please find my response below:
>
> **Q1:** The existence of a Nash Equilibrium makes sense. Limit on Search rounds is an important clarification. I could not find this detail in the paper. **Please consider adding this detail to the Training Details section (Section 4.1).**
>
> **Q2:** Please consider adding the details of answer set D to the main paper or specifically pointing towards the appendix where it is defined.
>
> **Q6:** Thank you for the clarification and new results on BrowseComp, but I would like to respectfully disagree with the first point here.
> - First, all the mentioned papers were released within a window of 2-3 months after the release of BrowseComp, making it difficult for authors of those works to incorporate a new benchmark. Note that any work released within a window of 3 months is considered a contemporary work. However, this paper is new, and authors should strive to report results on the most ecologically valid benchmark.
> - Second, I am not sure why using a live web retriever makes the comparison unfair? If you run all the baselines in your work using the same live web retriever with the same config, all the baseline results and proposed method results are comparable. Moreover, it is okay to use an in-house web retriever as long as the same retriever is used to compute the baseline numbers.
>
> **Q8:** Thank you for indulging my curiosity. Your explanation makes sense to me.
>
> Overall, I feel that this work is promising, and I do vote for acceptance of this work. However, the evaluation on classic Open-Domain QA benchmarks does not align with the framing of Deep Search Agents, when other benchmarks are purpose-built for testing deep search agents. Due to this, I cannot increase my score further.
>
> I am happy to engage in further discussions with the authors. Please feel free to correct me if my understanding is incorrect anywhere.

---

> > ### Author Response · Authors · 2025-11-28
> > **Response to Reviewer BnWv**
> >
> > Thank you very much for your thoughtful response and continued engagement with our work. We greatly appreciate your constructive feedback and the time you have dedicated to reviewing our paper.
> >
> > We are grateful for your specific suggestions regarding Q1 and Q2. We will incorporate the details about the limit on search rounds into the Training Details section (Section 4.1) and provide clearer references to the answer set D in the main paper, as you recommended. These clarifications will certainly improve the readability and completeness of our manuscript.
> >
> > Regarding Q6, we appreciate your perspective on the BrowseComp benchmark and acknowledge your valid points about ecological validity for deep search evaluation. We will carefully consider how to better position our evaluation framework in the revised version.
> >
> > We sincerely appreciate your positive assessment of our work and your vote for acceptance. Your feedback has been invaluable, and we are committed to addressing all the raised concerns in our revision to strengthen the paper.
> >
> > Thank you once again for your insightful comments and the constructive dialogue throughout this review process.

---

### Official Review · Reviewer_EH85 · 2025-11-01

**Soundness:** 3
**Presentation:** 3
**Contribution:** 3
**Rating:** 6
**Confidence:** 3

**Summary:**

The paper propose Search Self-Play (SSP), a self-play strategy to finetune LLMs to utilize multiturn search agents in response to a given question. The approach consists of training a single LLM to act both as task proposer and problem solver, in a zero-sum game fashion. The repeated game is constrained by making sure proposed tasks are meaningful and ground truth answers are correct via a RAG on the documents retrieved by the proposer’s trajectory.
Experiments performed on Qwen 7B illustrate the benefit of the approach, leading to substantial gains on Q&A benchmarks. Convincing ablations show the benefit of co-training both the proposer and solver objective, thus motivating the efficacy of the overall approach.

**Strengths:**

- The paper tackles the very relevant and challenging problem of improving LLMs for search and information retrieval.
- The zero-sum game formulation is very interesting but at the same time easily exploitable by both agents and thus I would have expect to produce degenerate solutions. Instead, the proposed constrained approach and overall meta-algorithm seems quite robust and able to produce stable improvements, as measured by the improved downstream performance during training.
- The experiments are sound and different design choices have been ablated, such as training only the solver or the proposer.

**Weaknesses:**

- I don’t understand why the proposer and solver are fine tuned with different update rules, REINFORCE and GRPO respectively. Couldn’t the same algorithm, say GRPO, be applied to both? This was not clear to me.
- Are both the proposer and solver updated at each step? I would expect more stable training dynamics to update the proposer less frequently. I would be curious to know the author’s view and experience on this. Also I think this would help understand the training dynamics.
- It would be nice to analyze the curriculum of tasks proposed by the proposer. How does it change compared to when it is not trained? Are they too hard, too easy, or easily hackable by the solver? Ultimately, this is at the core of the proposed approach.

**Questions:**

See weaknesses.

---

> ### Author Response · Authors · 2025-11-23
> **Response to Reviewer EH85(Part 1)**
>
> ### Q1: I don’t understand why the proposer and solver are fine tuned with different update rules, REINFORCE and GRPO respectively. Couldn’t the same algorithm, say GRPO, be applied to both? This was not clear to me.
>
> A1: We conducted comprehensive ablation studies to investigate exactly this question, and the results provide valuable insights into the trade-offs involved.
>
> As shown in the table below, using GRPO for both proposer and solver (GRPO--GRPO configuration) does achieve the highest average accuracy (50.9), slightly outperforming our default RF--GRPO configuration (49.5). **However, this improvement comes at a substantial computational cost: the per-step generation time increases from 83.4 seconds to 504.4 seconds—approximately 6× slower.** The performance gains are modest across most datasets (+1.2 on NQ, +2.0 on PopQA, +3.0 on 2Wiki, +1.2 on MuSiQue, +2.4 on Bamboogle), with even a small drop on TriviaQA (-0.6).
>
> | Proposer | Solver | NQ | TriviaQA | PopQA | HotpotQA | 2Wiki | MuSiQue | Bamboogle | Avg. | T/step(s) |
> | --- | --- | --- | --- | --- | --- | --- | --- | --- | --- | --- |
> | RF | GRPO | 54.8 | 73.4 | 51.8 | 51.8 | 38.8 | 21.2 | 54.4 | 49.5 | 83.4 |
> | RF | RF | 53.0 | 71.2 | 53.6 | 46.2 | 28.0 | 13.2 | 28.8 | 42.0 | 9.1 |
> | GRPO | RF | 51.2 | 69.2 | 49.8 | 47.6 | 37.0 | 21.6 | 48.8 | 46.5 | 50.1 |
> | GRPO | GRPO | 56.0 | 72.8 | 53.8 | 52.4 | 41.8 | 22.4 | 56.8 | 50.9 | 504.4 |
>
> *Table: Evaluation results and training time comparisons on different RL algorithms for proposers and solvers, where RF denotes REINFORCE.*
>
> Interestingly, when we examine other configurations, they all underperform compared to our default RF--GRPO setting. For instance, using GRPO for the proposer with RF for the solver leads to performance degradation (46.5), while the RF--RF pairing also shows inferior results (42.0). This demonstrates that our default configuration provides the optimal balance for the SSP framework.
>
> **Given that the GRPO--GRPO configuration requires 6× more computational resources for only marginal gains (1.4 points), the RF--GRPO combination provides the optimal trade-off between effectiveness and efficiency for practical training budgets**. This is why we chose it as our default configuration.
>
> ### Q2: Are both the proposer and solver updated at each step? I would expect more stable training dynamics to update the proposer less frequently. I would be curious to know the author’s view and experience on this. Also I think this would help understand the training dynamics.
>
> A2: Yes, we do update both the proposer and solver at each training step. However, your intuition about less frequent proposer updates is actually addressed by our replay buffer mechanism, which creates an **effective frequency imbalance** between the two roles. Here's how our replay buffer achieves this:
>
> - **Proposer**: Generates new valid questions that enter the replay buffer, but these questions are used across multiple training iterations before being cleared (every 10 steps in our periodic reset strategy)
> - **Solver**: Trains on both newly generated questions AND previously generated questions from the replay buffer, effectively seeing more training examples per step
>
> This design creates a natural imbalance where the solver receives **more training signal per step** compared to the proposer, which is functionally equivalent to updating the proposer less frequently as you suggested. The replay buffer allows the solver to learn more thoroughly from each question the proposer generates, improving data utilization efficiency and enabling the solver to fully internalize the knowledge from the proposer's carefully crafted problems.
>
> This mechanism addresses your concern about training stability: the solver gets multiple opportunities to learn from high-quality questions while the proposer's updates remain consistent, leading to the stable co-evolution dynamics we observe in Figure 3. The replay buffer essentially acts as a natural regularizer that prevents rapid oscillations between the two roles while maximizing learning efficiency.

---

> > ### Author Response · Authors · 2025-11-23
> > **Response to Reviewer EH85(Part 2)**
> >
> > ### Q3 It would be nice to analyze the curriculum of tasks proposed by the proposer. How does it change compared to when it is not trained? Are they too hard, too easy, or easily hackable by the solver? Ultimately, this is at the core of the proposed approach.
> >
> > A3: We completely agree that analyzing the proposer's curriculum evolution is crucial to understanding the effectiveness of SSP. We have conducted a thorough analysis of how the proposer's task curriculum evolves during training, which we present in newly-added Figure 6 in the Appendix. This analysis reveals four key dimensions of curriculum development:
> >
> > **Progressive Difficulty Scaling (Figure 6c):** To systematically track question difficulty evolution, we employed DeepSeek-V3.2 to evaluate generated questions using a structured 5-point difficulty scale (detailed in Appendix). The results show a clear upward trend in question difficulty throughout training, with average difficulty scores increasing from 2.78 to 3.10. This demonstrates that the proposer successfully learns to generate progressively more challenging questions rather than remaining static or becoming too easy for the solver.
> >
> > **Enhanced Exploration Capabilities (Figure 6a):** The proposer gradually increases its search tool usage which indicates that the trained proposer develops more sophisticated information-gathering strategies, creating questions that require deeper exploration rather than simple factual recall.
> >
> > **Quality Improvement Through Training (Figure 6b):** The question validation success rate reveals a critical learning progression. Initially, the proposer struggles with proper question formatting and generation protocols, resulting in very low pass rates (near 0%). As training progresses, the proposer learns to follow correct formatting guidelines and generate well-structured questions, with the validation success rate steadily improving to approximately 50%. This demonstrates that the proposer evolves from producing mostly invalid outputs to generating high-quality, properly formatted questions that meet verification standards.
> >
> > **Balanced Topic Coverage (Figure 6d):** The proposer maintains balanced topic distribution across 8 major categories throughout training. This prevents the proposer from exploiting narrow patterns or generating easily hackable questions that could be solved through memorization or simple heuristics.
> >
> > This comprehensive curriculum analysis confirms that the proposer learns to generate an adaptive, progressively challenging, and diverse set of tasks that effectively drive the co-evolutionary improvement central to our approach.

---

> ### Comment · Area_Chair_GuGk · 2025-11-25
> **Please participate in discussions with authors and other reviewers asap**
>
> Please ensure you are actively participating in the discussion with authors.
>
> Additionally, I strongly encourage you to read the other reviews and discuss with your fellow reviewers. It is vital to compare perspectives and raise any remaining concerns now to give the authors a fair opportunity to respond.
>
> Based on these interactions, please update your reviews and finalize your decisions.
>
> Best, AC

---

### Author Response · Authors · 2025-12-01
**Summary to New AC**

We thank the ACs for the guidance and support throughout the review process. We also would like to express our sincere gratitude to all the reviewers for their time, insightful suggestions, and valuable comments. We deeply appreciate the positive recognition from the reviewers regarding our paper's novel methodology (EH85, BnWv, TBC9, tjcn), thorough experiments (EH85, BnWv, TBC9, tjcn), and clear presentation (TBC9). In the discussion period, we have provided detailed responses to each specific question raised by the reviewers. Below is a summary of our key responses:

- For reviewer BnWv and tjcn, we clarified the construction and usage of the pre-defined answer set $D$, adding details to Section 4.1 and Appendix A.4.
- For reviewer TBC9 and tjcn, we have conducted EM/F1 evaluations on all relevant benchmarks.
- For reviewer EH85, we conducted comprehensive ablation studies to investigate different RL algorithms for proposers and solvers.
- For reviewer EH85, we conducted a thorough analysis of how the proposer's task curriculum evolves during training and added figures to Appendix.
- For reviewer BnWv, TBC9 and tjcn, we performed additional evaluation on the BrowseComp benchmark using an in-house online web retriever and reported the results.

We believe our responses effectively address the reviewers' concerns. Once again, thank you for your careful consideration. We appreciate the time and effort devoted by reviewers and ACs.

---

### Meta-Review · Area_Chair_y9ST · 2026-01-07

**Summary:**

This paper proposes Search Self-Play (SSP), a reinforcement learning framework for training deep search agents without human supervision. The approach uses a single LLM in dual roles: a task proposer that generates search-intensive questions and a problem solver that answers them through multi-turn search interactions. A RAG-based verification mechanism ensures the validity of questions.

The reviewers generally found the approach novel and well-motivated, with strong empirical results across seven QA benchmarks. Key concerns centered on: (1) algorithm design choices, (2) training stability and computational costs, (3) limited evaluation on newer benchmarks like BrowseComp, (4) reliability of the verification mechanism, and (5) details about the answer set construction.

The authors provided comprehensive rebuttals with additional ablations, curriculum analysis, and supplementary experiments, addressing most reviewer concerns. The remaining problems are relatively minor and can be addressed in the camera-ready version.

**Suggested revisions for camera-ready:**

- Expand discussion on the high-precision verifier design choice and its tradeoffs.
- From the AC’s reading of the paper, it is necessary to discuss the performance gap between training Qwen2.5-7B from scratch with SSP versus Search-R1, and whether self-play is more suited for continual training than training from scratch.
- Include BrowseComp results with appropriate caveats about reproducibility, as pointed out by multiple reviewers.

**Reviewer Concerns:**

**Addressed concerns:**

- Algorithm design choices (EH85, TBC9): The authors provided extensive ablations demonstrating the efficiency-performance tradeoff, justifying their default configuration.
- Training dynamics and stability (EH85, TBC9): The authors clarified the replay buffer mechanism and provided additional figures showing stable co-evolution dynamics.
- Curriculum analysis (EH85, BnWv): New results demonstrate progressive difficulty scaling, enhanced exploration, and balanced topic coverage.
- Computational cost breakdown (TBC9, BnWv): The authors reported resource usages.
- Answer set composition (BnWv, tjcn): The authors clarified the construction process.

**Outstanding concerns:**
- BrowseComp evaluation (BnWv, tjcn): Authors provided results using an in-house retriever but could not provide directly comparable baseline numbers due to reproducibility constraints. Reviewer BnWv explicitly acknowledged this limitation but still voted for acceptance.
- Verifier reliability (tjcn): The authors justified the high-precision, lower-recall design choice, but additional clarification on this tradeoff would strengthen the paper.

**Reviewer Scores:**

**EH85 (6):** Likely unchanged or slight increase. All concerns were thoroughly addressed. No post-rebuttal response.

**BnWv (6):** Likely unchanged or slight increase. The reviewer explicitly stated "I do vote for acceptance" after the rebuttal, though noted the evaluation framing concern remains.

**TBC9 (6):** Likely unchanged or slight increase. Comprehensive responses provided. No post-rebuttal response.

**tjcn (4):** Likely slight increase. Most concerns were addressed. The verifier's reliability concern received a reasonable justification, though further elaboration would help. No post-rebuttal response.

---

### Decision · Program_Chairs · 2026-01-26

Accept (Poster)